# Language Model Tokenizers Introduce Unfairness Between Languages

**Aleksandar Petrov, Emanuele La Malfa, Philip H.S. Torr, Adel Bibi**
University of Oxford
aleks@robots.ox.ac.uk

## Abstract

Recent language models have shown impressive multilingual performance, even when not explicitly trained for it. Despite this, there are concerns about the quality of their outputs across different languages. In this paper, we show how disparity in the treatment of different languages arises at the tokenization stage, well before a model is even invoked. The same text translated into different languages can have drastically different tokenization lengths, with differences up to 15 times in some cases. These disparities persist even for tokenizers that are intentionally trained for multilingual support. Character-level and byte-level models also exhibit over 4 times the difference in the encoding length for some language pairs. This induces unfair treatment for some language communities in regard to the cost of accessing commercial language services, the processing time and latency, as well as the amount of content that can be provided as context to the models. Therefore, we make the case that we should train future language models using multilingually fair subword tokenizers.

## 1 Introduction

Language models are increasingly important in natural language processing tasks, as they can understand and generate human-like language. They have been deployed in applications such as virtual assistants (Chen et al., 2021; Ouyang et al., 2022), chatbots (Kuhail et al., 2023; Lee et al., 2023), machine translation (Stahlberg, 2020; Ranathunga et al., 2023), and text summarization (Kryściński et al., 2019; Xu et al., 2020). As general-purpose technologies, it is also projected that Large Language Models (LLMs) will have a significant impact on the economy and the labour market (Teubner et al., 2023; Eloundou et al., 2023).

Such LLMs are often trained using large swaths of internet content regardless of language. Hence, these models often end up being multilingual, even if not by design. ChatGPT (OpenAI, 2022) is a prominent recent example (Bang et al., 2023; Jiao et al., 2023; Johnson, 2023). Given the economic benefits of LLMs and LLM-derived technology, it's beneficial that they support multiple languages. Equal access is crucial, and multilingual support is a key component of this.

However, this multilingualism is currently treated as a curious emergent phenomenon rather than a carefully designed, controlled and managed process. The performance of LLMs has been shown to be generally lower in non-target languages, a problem especially pronounced for low-resource languages (Virtanen et al., 2019; Ahuja et al., 2023). Providing access to the same technology in different languages but moderation and safety tools only for some has resulted in dire societal consequences before (Stecklow, 2018; Facebook, 2021; Leung, 2022). Differing cost of access could also reinforce inequality in opportunities for economic mobility and social participation (Lythreatis et al., 2022). Therefore, as LLM multilingualism emerges,

37th Conference on Neural Information Processing Systems (NeurIPS 2023).

we should pay attention to ensuring comparable performance and accessibility across the supported languages, regardless of whether by design or by chance.

This work demonstrates how the unequal treatment of languages arises at the tokenization stage,[1] well before the language model sees any data at all. For instance, the tokenizer employed by ChatGPT (OpenAI, 2022) and GPT-4 (OpenAI, 2023) uses about 1.6 times more tokens to encode the same text in Italian as it does in English, 2.6 times for Bulgarian and 3 times for Arabic. For Shan —the native language of people from the Shan State in Myanmar— that difference can be as high as 15 times. Unicode character and byte-level tokenization also result in drastically different encoding lengths across languages: byte-level representation of the same text is over 4 times longer for Burmese or Tibetan than Chinese.

We discuss three fairness implications of these differences in tokenization:

1. **Cost:** Commercial services charge users per token or Unicode character. In either case, these discrepancies lead to users of some languages paying at least 2.5 times more for the same task as users of English.
2. **Latency:** The number of tokens has a direct effect on the processing time for a task. Some languages can require twice the time to process the same content as English. This may be critical for real-time applications like emergency services.
3. **Long context processing:** Many models have a fixed-size context. Users of languages that are more token-efficient can use these systems to process or generate texts that may be more than an order of magnitude longer than users of other languages. This may lead to significant discrepancies in the quality of service.

Therefore, we make the case for *multilingual tokenization parity*: tokenizers should produce similar encoded lengths for the same content across languages. Hence, we advocate for multilingually fair tokenizers for the next generation of language models.

## 2 Intriguing Properties of Tokenization Across Languages

Subword tokenization is currently the preferred approach for state of the art language models (Kudo and Richardson, 2018). In this section, we show how artefacts from data collection might result in technical terms or rare words having dedicated tokens, while more commonly used words and non-Latin characters end up requiring multiple tokens.

Using large corpora scraped from the internet results in *peculiar* choices for tokens. For instance, GPT-2 contains *glitch tokens* which can be usernames or concepts from games (Rumbelow and Watkins, 2023b; Miles and Riley, 2023). As an example, `BuyableInstoreAndOnline`, likely coming from an online store backend, has a dedicated token. Another such token is `rawdownloadcloneembedreportprint`.

While such obscure terms get their own tokens, the frequently used Arabic word "لماذا" (meaning "why") is broken into letters with each letter having its own token. The same word in Bulgarian ("защо") is not only broken down to letters, but some of the let-ters require two tokens to be represented, resulting in 6 tokens for this 4 letter word.

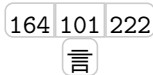

One may argue that this is because Arabic and Bulgarian are not target languages for GPT-2. However, glitch tokens also exist for Japanese: there are dedicated tokens for "ゼウス", the name of the ancient Greek god Zeus and "サーティワン", the name of an ice cream chain (Rumbelow and Watkins, 2023a). At the same time, GPT-2 requires 3 tokens to represent the much more commonly used kanji character for "to say":

In fact, more than half of the Japanese kanji characters require three tokens.

---

[1]We offer a summary of the relevant tokenization approaches in Appendix A.

The existence of glitch tokens like "ゼウス" and "サーティワン" despite the lack of a dedicated token for "言" shows that tokenizers are heavily influenced by the biases of the corpus source. If one uses non-natural inputs, log files, or specialist forums, the tokenizer vocabulary would reflect this. While `cl100k_base`, the tokenizer used for the newer ChatGPT and GPT-4, may not have glitch tokens it still requires two tokens to represent some Cyrillic letters and three tokens for more than 65% of kanji characters. Therefore, to place all languages on an equal footing, it is important to have the tokens balanced across languages.

## 3 Measuring Tokenizer Parity

To demonstrate that the above examples are not anecdotal evidence, we introduce the notion of *tokenizer parity* to systematically assess how fairly tokenizers treat equivalent sentences in different languages. Parity occurs when a tokenizer exhibits similar tokenized lengths for the same sentence in different languages. Take a sentence $s_A$ in language $A$ and its translation $s_B$ to language $B$. Then, a tokenizer $t$ achieves parity for $A$ with respect to $B$ at $s_A$ and $s_B$ if $|t(s_A)|/|t(s_B)| \approx 1$, where $t(s_A)$ is the tokenization of the sentence $s_A$ and $|t(s_A)|$ represents its length. The ratio $|t(s_A)|/|t(s_B)|$ is the *premium* for $A$ relative to $B$. [2]

## 4 Tokenization Length Differences Across Languages

Languages vary significantly in the number of tokens required to encode the same content, as demonstrated in the examples in Section 2. Hence, following Section 3, we measure the tokenization premium of different tokenizers. To this end, we use the FLORES-200 parallel corpus, comprising of the same 2000 sentences taken from Wikipedia and human-translated to 200 different languages (Guzmán et al., 2019; Goyal et al., 2021; Costa-jussà et al., 2022). We look at subword tokenization models which target English, languages other than English, language varieties, multi-lingual tokenizers, as well as tokenizer-free (byte-level) modelling.

### 4.1 Parity for English-centric Models

As most models target English, we report in Table 1 the tokenization parity for a subset of languages in FLORES-200. The parities for all 200 languages are in Appendix C. [3] GPT-2 (Radford et al., 2019), RoBERTa (Liu et al., 2019), and the `r50k_base`, `p50k_base` and `p50k_edit` tokenizers (OpenAI, 2022) have close[4] tokenization lengths so we report them together. ChatGPT and GPT-4 share the same `cl100k_base` tokenizer and are also reported together. Some models, such as FlanT5 (Chung et al., 2022), use a special `UNK` token to model unknown symbols not encountered during training. Hence, to ensure a fair comparison, we report only languages where no more than 10% of the input characters are mapped to `UNK` tokens (marked with —).

Table 1 shows large variations in the tokenizer parity for all tokenizers. For GPT-2 and RoBERTa, Pangasinan, the language with shortest tokenization, is already 66% more expensive to process than English. ChatGPT and GPT-4 are slightly closer to parity, likely

Table 1: Premiums with respect to English on FLORES-200 for several **English-centric** models. The languages in the top or bottom three for any tokenizer, as well as the ones discussed in the text, are shown.

| | GPT-2 RoBERTa | ChatGPT GPT-4 | FlanT5 |
|---|---|---|---|
| Bulgarian | 5.51 | 2.64 | — |
| Burmese | 16.89 | 11.70 | — |
| Chinese (Simplified) | 3.21 | 1.91 | — |
| Dzongkha | 16.36 | 12.33 | — |
| English | 1.00 | 1.00 | 1.00 |
| French | 2.00 | 1.60 | 1.60 |
| German | 2.14 | 1.58 | 1.37 |
| Italian | 2.01 | 1.64 | 2.18 |
| Japanese | 3.00 | 2.30 | — |
| Jingpho | 2.65 | 2.35 | 3.41 |
| Maori | 2.45 | 2.35 | 3.28 |
| Norwegian Bokmål | 1.86 | 1.56 | 2.24 |
| Odia | 13.38 | 12.48 | — |
| Pangasinan | 1.66 | 1.57 | 2.18 |
| Portuguese | 1.94 | 1.48 | 2.21 |
| Romanian | 2.48 | 1.88 | 1.50 |
| Santali | 12.86 | 12.80 | — |
| Shan | 18.76 | 15.05 | — |
| Spanish | 1.99 | 1.55 | 2.23 |
| Standard Arabic | 4.40 | 3.04 | — |
| Tumbuka | 2.78 | 2.57 | 3.29 |
| Vietnamese | 4.54 | 2.45 | — |

[2] The concurrent work by Ahia et al. (2023) also evaluates the tokenization premiums for different languages and reaches similar conclusions.

[3] An interactive table of all the languages and tokenizers is also available on the project website.

[4] The largest tokenizer parity difference between them is less than 0.005.

Table 2: Tokenizer premiums on the FLORES-200 dataset for **non-English centric models**. The premium is computed with respect to the target language (Modern Standard Arabic was used for Arabic BERT and Simplified Chinese for RoCBert). The languages that are in the top or bottom two for any tokenizer as well as the ones discussed are shown.

| | Arabic BERT | RoCBert (Chinese) | CamemBERT (French) | GottBERT (German) | BERT Japanese | PhoBERT (Vietnamese) |
|---|---|---|---|---|---|---|
| Belarusian | 4.74 | — | — | 5.62 | — | 3.46 |
| Bulgarian | 4.30 | — | — | 4.73 | — | 3.09 |
| Catalan | 2.36 | 2.86 | 1.59 | 1.89 | 1.95 | 1.57 |
| Chinese (Simp.) | — | 1.00 | — | 3.95 | 0.82 | — |
| Chinese (Trad.) | — | 0.94 | — | 3.82 | 0.84 | — |
| Dutch | 2.52 | 2.92 | 1.68 | 1.73 | 1.98 | 1.58 |
| Dzongkha | — | — | — | 16.12 | — | — |
| English | 1.83 | 2.60 | 1.20 | 1.35 | 1.49 | 1.20 |
| French | 2.42 | 3.10 | 1.00 | 1.99 | 2.03 | 1.66 |
| Friulian | 2.33 | 2.79 | 1.66 | 1.98 | 1.92 | 1.59 |
| German | 2.63 | 3.12 | 1.85 | 1.00 | 2.04 | 1.67 |
| Greek | 4.93 | 3.00 | — | 6.73 | — | 3.73 |
| Italian | 2.58 | 3.10 | 1.63 | 1.93 | 2.04 | 1.60 |
| Japanese | 1.85 | 1.34 | — | 4.35 | 1.00 | — |
| Jingpho | 3.12 | 3.12 | 2.13 | 2.55 | 2.47 | 1.84 |
| Luxembourgish | 2.56 | 2.97 | 1.82 | 1.75 | 1.96 | 1.72 |
| N. Lev. Arabic | 1.00 | — | — | 6.52 | — | — |
| Shan | — | — | — | 16.88 | — | — |
| Standard Arabic | 1.00 | — | — | 7.03 | — | — |
| Tagalog | 2.84 | 3.28 | 2.00 | 2.20 | 2.39 | 1.74 |
| Tosk Albanian | 2.66 | 2.90 | 2.17 | 2.39 | — | 2.02 |
| Tsonga | 3.01 | 3.09 | 2.03 | 2.29 | 2.46 | 1.76 |
| Tumbuka | 3.27 | 3.49 | 2.21 | 2.61 | — | 2.00 |
| Vietnamese | 2.52 | 2.55 | — | 4.12 | — | 1.00 |
| Yue Chinese | — | 0.92 | — | 3.75 | — | — |

Table 3: Tokenizer premiums on the FLORES-200 dataset for the MuRIL model focusing on **16 Indian languages and English**. The premium is computed with respect to English.

| | MuRIL |
|---|---|
| English | 1.00 |
| Nepali | 1.01 |
| Bengali | 1.01 |
| Tamil | 1.06 |
| Marathi | 1.06 |
| Kannada | 1.06 |
| Hindi | 1.16 |
| Malayalam | 1.18 |
| Gujarati | 1.19 |
| Sanskrit | 1.21 |
| Telugu | 1.21 |
| Odia | 1.21 |
| Sindhi | 1.22 |
| Assamese | 1.24 |
| Urdu | 1.26 |
| Eastern Panjabi | 1.35 |
| Kashmiri (Arabic) | 1.75 |
| Kashmiri (Devanagari) | 1.75 |

due to their larger vocabulary size. However, the cheapest languages, Portuguese, Pangasinan and German, still see a premium of 50% when compared to English. Shan has the worst tokenizer parity for all four models. Take as an example "ၷ̇ː", one of the Shan words for "you". It is tokenized by ChatGPT and GPT-4 as:

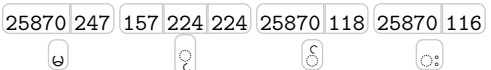

This word is constructed from one consonant and three diacritics. As the diacritics are encoded separately, there are four Unicode codepoints for this Shan character, resulting in 9 tokens. The English "you" has three characters but a single token.

FlanT5 has more than 10% `UNK` tokens for 42% of languages (— in Table 1). It has a higher premium than the other tokenizers for all other languages except German and Romanian.

**Summary.** All four English-centric tokenizers we consider are far from tokenization parity. Portuguese is closest to parity with English for the ChatGPT and GPT-4 tokenizer but still requires about 50% more tokens for the same content. Shan is furthest from parity for this tokenizer with 15 times longer encodings compared to English. FlanT5 is closer to parity with its premium range 1.37–3.41 but it encodes only 54% of the languages, so we cannot say that it is more multilingually fair than the other tokenizers.

## 4.2 Parity for Models with Other Target Languages

There are models targeting languages other than English as well. Table 2 shows six such models based on the BERT architecture (Devlin et al., 2019): ArabicBERT (Safaya et al., 2020), RoCBert for Chinese (Su et al., 2022), CamemBERT for French (Martin et al., 2020), GottBERT for German (Scheible et al., 2020), BERT Japanese (Tohoku NLP Group, 2019) and PhoBERT for Vietnamese (Nguyen and Nguyen, 2020).

Table 4: Tokenizer premiums with respect to English on FLORES-200 for **multilingual models**. The languages that are in the top or bottom two for any tokenizer, as well as the ones discussed in the text, are shown.

| | XLM-R | NLLB | mT5 | M2M100 | BLOOM |
|---|---|---|---|---|---|
| Bulgarian | 1.16 | 1.31 | 1.28 | 1.23 | 2.49 |
| Central Kanuri | 2.60 | 2.54 | 2.43 | 2.49 | 2.10 |
| Chinese (Simp.) | 0.97 | 1.11 | 0.92 | 1.05 | 0.95 |
| Dzongkha | — | 1.48 | 4.24 | — | 7.36 |
| English | 1.00 | 1.00 | 1.00 | 1.00 | 1.00 |
| Indonesian | 0.94 | 0.93 | 1,08 | 0.98 | 0.96 |
| Italian | 1.19 | 1.25 | 1.34 | 1.25 | 1.62 |
| Japanese | 1.11 | 1.01 | 0.90 | 1.20 | 1.81 |
| Kabiyè | 2.98 | 1.56 | 2.83 | 2.71 | 3.34 |
| Santali | — | 2.49 | — | — | 12.71 |
| Shan | 4.43 | 1.94 | 3.28 | 4.63 | 12.06 |
| Std. Arabic | 1.18 | 1.40 | 1.35 | 1.29 | 1.14 |
| Std. Tibetan | — | 1.44 | 3.68 | — | 6.66 |
| Uyghur | 1.41 | 1.40 | 2.57 | 3.00 | 3.67 |
| Yue Chinese | 0.93 | 1.05 | 0.95 | 1.03 | 0.93 |

Table 5: Tokenizer premiums with respect to English on FLORES-200 for **byte-level models**. The languages that are in the top or bottom two for any tokenizer, as well as the ones discussed in the text, are shown.

| | CANINE UTF-32 bytes | ByT5 UTF-8 bytes |
|---|---|---|
| Bulgarian | 1.04 | 1.89 |
| Burmese | 1.24 | 3.51 |
| Chinese (Simplified) | 0.34 | 0.93 |
| Chinese (Traditional) | 0.32 | 0.89 |
| Dzongkha | 1.25 | 3.64 |
| English | 1.00 | 1.00 |
| Italian | 1.18 | 1.19 |
| Japanese | 0.44 | 1.27 |
| Shan | 1.42 | 3.94 |
| Standard Arabic | 0.88 | 1.60 |
| Standard Tibetan | 1.13 | 3.31 |
| Tok Pisin | 1.28 | 1.28 |
| Tumbuka | 1.30 | 1.32 |
| Yue Chinese | 0.31 | 0.87 |

The English premium for GottBERT (1.35) is lower than those for Dutch (1.73) and Luxembourgish (1.75), which are more linguistically similar to German. CamemBERT is similar: English has the lowest premium (1.20), while Catalan (1.59) and Friulian (1.66) have higher premiums. PhoBERT also has English with the lowest tokenizer premium (1.20). Thus, even models targeting other languages exhibit a preference for English tokenization.

RoCBert and BERT Japanese differ by having the other target language as the one closest to parity, possibly due to the partially shared script. ArabicBERT demonstrates a similar behaviour, with Central Kanuri (1.27) and Acehnese (1.73), both written in Arabic script, and with English at 1.82. Sharing writing systems seems to improve tokenization parity.

Across all tokenizers, the premium for English relative to the respective target language is significantly lower than the premium of RoBERTa for that target language. This asymmetry between English and all other languages likely stems from the extensive incorporation of English in documents written in other languages (Zhang et al., 2022).

We also consider MuRIL, a BERT-based model trained on 16 Indian languages and English (Khanuja et al., 2021). Despite the model's focus on Indian languages, it remains most token-efficient for English (see Table 3).

Unequal treatment of dialects or linguistic varieties can lead to social and economic disadvantages making it important to also study the tokenization differences between the "standard" language and its varieties. For Swiss German and the Mauritian and Haitian Creoles, there are large differences in tokenization lengths compared respectively to High German (on GottBERT) and French (on CamemBERT). English is much closer to parity for both models than these language varieties. Therefore subword tokenizers might not be able to generalize to language varieties, such as dialects and creoles. The tokenizers of ArabicBERT and BERT Japanese, however, are close to parity across various dialects of both languages and have lower premiums for the dialects than for English. This is likely due to the good representation of the dialects in the dataset as well as the dialects being linguistically closer to the respective standard languages. The detailed analysis is deferred to Appendix B.

**Summary.** We observed that the tokenizers targeting French, German and Vietnamese have English as the language closest to parity, rather than more linguistically close languages. On the other hand, tokenizers for Arabic, Chinese and Japanese have lower premiums for languages they share a script with. Notably, despite targeting Indian languages, MuRIL still has the shortest tokenizations for English. Finally, across all tokenizers, the premium for English is lower than the premium for the same language for the English-centric RoBERTa. Hence, we conclude that tokenizers for other languages give English preferential treatment.

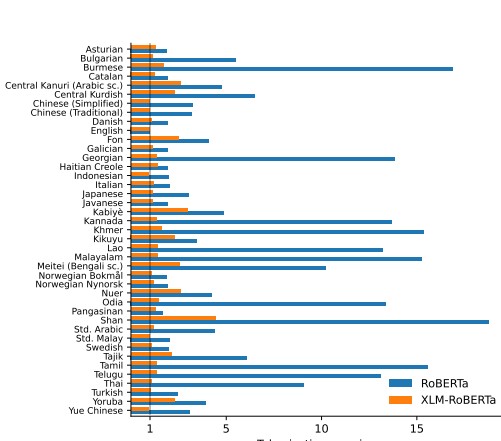

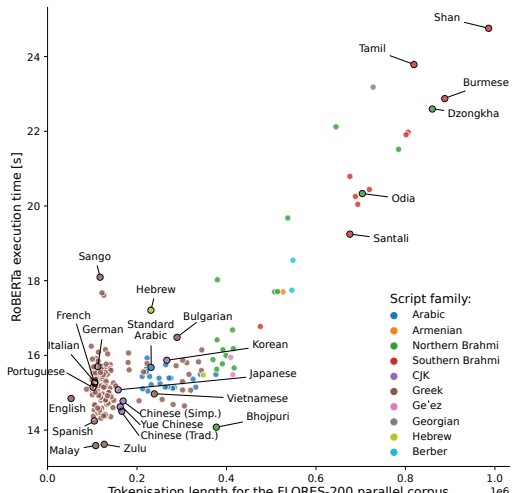

Figure 1: Comparison of the tokenization premiums for XLM-R and RoBERTa for the subset of languages that XLM-R encodes with less than 10% to the UNK token.

Figure 2: Average processing time and length of the tokenized inputs of RoBERTa. Each FLORES-200 sentence is processed for 20 independent runs. The script family designation is only for illustration purposes.

## 4.3 Parity for Multilingual Models

There has been a growing interest in multilingual language models, particularly for translation (Dabre et al., 2020). As these models are intended to support a variety of languages, one would expect them to be close to tokenizer parity. We compare several such multilingual models: XML-R (Conneau et al., 2020), NLLB (Costa-jussà et al., 2022), M2M100 (Fan et al., 2021) and mT5 (Xue et al., 2020). All of these models use the SentencePiece tokenizer with upsampling for rare languages. The final model, BLOOM (Scao et al., 2022), uses byte-level BPE instead of SentencePiece and is designed to maintain similar ratios of tokens per word for each language as reference monolingual tokenizers.

BLOOM and NLLB encode all languages with less than 10% UNK tokens, respectively thanks to byte-level BPE tokenization and being trained on the same 200 languages as FLORES-200 (see Table 4). The other three models fail to encode at least one language. All five models have languages with premiums of more than 2.5. Still, all models are better than the English-centric models in Table 1. Figure 1 shows how XLM-R is much closer to parity than RoBERTa (on which it is based), over all languages it can encode. However, none of the models uniformly reaches parity across all languages. Therefore even models which are intentionally designed to be multilingual suffer from a lack of tokenization parity.

**Summary:** Multilingual models can improve the tokenization parity for different languages but challenges remain in achieving tokenization parity across all languages.

## 4.4 Parity for Byte-level Tokenization Models

Byte-level representation is crucial for multilingual support, as it encodes any Unicode codepoint, even if unseen during training. One can also bypass vocabulary construction and directly employ the 256 byte values, enabling end-to-end training (*byte-level tokenization*). CANINE (Clark et al., 2022) is a large model that operates at the Unicode codepoint level rather than the byte level. The CANINE tokenizer is thus equivalent to the UTF-32 encoding, resulting in an implicit tokenizer with a vocabulary of 1,114,112. ByT5 (Xue et al., 2022), on the other hand, uses the UTF-8 encoding: an implicit vocabulary of 256 tokens.[5]

---

[5]To be consistent, we will refer to the characters and bytes in the encoding of the CANINE and ByT5 tokenizers as *tokens* as they fulfil a similar role.

These byte-level models can represent any Unicode codepoint without an explicit tokenization step but there are still significant tokenization disparities. For CANINE, Shan has a premium of 4.58 relative to Yue Chinese. This can be attributed to the fact that CANINE provides a single token for each Unicode codepoint, which results in Chinese being more token-efficient (with a premium range 0.31–0.34 relative to English for the three Chinese languages) as each character is treated as a single token. This encoding also puts Shan at a disadvantage, as its encoding relies on diacritics represented as separate Unicode codepoints. Other languages, such as Tok Pisin and Tumbuka, which use the Latin script but require more characters than English for the same text, also face similar challenges.

Tokenization disparity is also present in the ByT5 model. The tokenization premium for ByT5 ranges from 0.87 (for Yue Chinese) to 3.94 (for Shan). The introduction of the variable-width UTF-8 encoding of Unicode characters in ByT5 creates another issue of unequal treatment. ASCII characters, which are sufficient for English, require only one byte. Other Latin script characters, as well as Greek, Cyrillic, Coptic, Armenian, Hebrew, Arabic and Syriac, require two bytes, while Chinese, Japanese and Korean characters require three bytes. Therefore, the tokenization of Chinese and Japanese is about three times as long for ByT5 as it is for CANINE (Table 5). Shan's premium of 3.94 is due to the fact that all its consonants and diacritics require three bytes. For example, the word "ၶႂ်ႈ" is encoded by ByT5 as 12 tokens, whereas the corresponding "you" requires 3 tokens. The situation is similar for other languages like Dzongkha, Tibetan and Burmese.

**Summary.** Byte-level models also fail to achieve parity among the languages from FLORES-200 exhibiting a premium of over 4 times for some language pairs. There are two sources of multilingual tokenizer disparities. First, there are natural differences in the number of characters used in different languages to communicate the same content. Second, the UTF-8 standard uses different number of bytes to encode codepoints of different scripts.

## 5  Fairness Implications of Tokenization Length Differences

We showed that no matter whether one uses subword, multilingual, or byte-level tokenization, none of the tokenizers gets close to parity for all languages in FLORES-200. This lack of tokenization parity is not merely a curiosity: it leads to unfairness in the cost to access language models, the latency of the service and the amount of data that can be processed.

### 5.1  Cost

It is increasingly common to access LLMs as paid API services. One pricing approach, employed by OpenAI at the time of writing,[6] is to charge per token. Therefore, the tokenization premiums discussed in Section 4 directly map to cost premiums. For ChatGPT and GPT-4, the cost to process a text in German or Italian is about 50% higher than to process the same text in English (Table 1). Using them in Dzongkha, Odia, Santali or Shan, the most expensive languages for these services, costs more than 12 times more than in English.

Another pricing strategy is per Unicode character: the approach currently taken by the Google Cloud Natural Language service.[7] However, as we showed in Section 4.4, the same content can have very different lengths when measured in Unicode characters. Burmese, Dzongkha, Shan, Tok Pisin or Tumbuka require more than 4 times more characters than Yue Chinese for the same text, resulting in a proportional cost difference. Therefore, both the per-token and the per-character approaches result in large disparities in the cost for users of different languages to use the exact same service.

### 5.2  Latency

High latency of real-time interactions for users of certain languages can result in a suboptimal experience and communication breakdowns. For customer support or emergency services, delays in response time can lead to miscommunication or delayed assistance.

---

[6] https://openai.com/pricing
[7] https://cloud.google.com/natural-language/pricing

As some languages have significantly longer tokenized inputs, they would also experience longer processing times. The transformer attention mechanism has a quadratic complexity in the number of input tokens (Keles et al., 2023). However, the full model architecture contains other submodules and therefore the overall complexity might be different.

To assess the effect of the tokenization length on the latency, in Figure 2 we plot the computation time of RoBERTa against the tokenization lengths. It appears that the processing time is linear in the tokenization length rather than quadratic, showing a strong correlation between sequence length and execution time. Therefore, tokenization disparities across languages also affect the latency and processing time for text in these languages.

As expected, English is on the left lower corner, having the shortest tokenization and one of the fastest processing times. Shan is on the other extreme with the longest tokenization length and execution time (almost twice that of English). We can also observe clear trends dependent on the script used. Latin script and other Greek-derived scripts show the shortest tokenization lengths and processing times followed by the Chinese-Japanese-Korean (CJK) and Arabic languages. Other predominantly Asian and African scripts have longer tokenization lengths and processing times.

The latency implications of tokenization disparity are not limited to text models. Speech recognition models often produce a series of tokens as their output sequentially. Similarly, speech synthesis takes as an input tokenized text (Latif et al., 2023). Therefore, differences in tokenization affect speech models too.

### 5.3 Long context processing

Transformers models have difficulty processing long inputs (Liu et al., 2023). Given that the size of the input is contingent upon the tokenization process, inputs of greater length may impose a challenge for language models to adequately reason over. Such a predicament may result in reduced abilities or limited applicability for languages with high tokenization premiums. For example, RoBERTa has a fixed block size of 512, GPT-2 has 768, 1024, 1280, or 1600 Radford et al. (2019), GPT-4 comes in 8,000 and 16,000 context variants.[8] These models cannot process inputs longer than that. Therefore, one can process less than a tenth of the content in languages like Burmese and Dzongkha than they can in English.

Alongside inconveniencing the users of these languages, this can also result in diminished performance on automated systems, such as content moderation. Reliable content moderation is crucial for tackling hate speech and diminished performance has already been shown to fail to prevent its spread (Stecklow, 2018; Facebook, 2021). Therefore, reduced long context capabilities for some languages could have severe real-world impacts.

## 6  Towards Multilingual Tokenization Fairness

Section 5 showed that high values of tokenization parity for a language lead to increased cost and latency and decreased capacity for long context processing. In this section, we argue that training language models from scratch with a multilingually fair subword tokenizer is the only approach that can effectively address all these aspects of tokenization unfairness.

**Subword tokenization is necessary to achieve parity.**  In Section 4.4, we showed that neither character-level nor byte-level input representation can achieve tokenization parity. Therefore, a variation of subword tokenization is necessary. For example, Chinese characters could be individual tokens, Latin characters might be represented as tokens with an average length of about 3 characters while pairs of Burmese characters and their diacritics being assigned single tokens. Such an approach would account for Chinese requiring one-third the characters English does (as shown in Table 5).

**A separate tokenizer for determining the processing cost is not sufficient.**  An easy patch for existing models is to use a separate tokenizer for calculating how much a user should be charged. Using one tokenizer for computing the cost and another to process

---

[8] https://openai.com/pricing

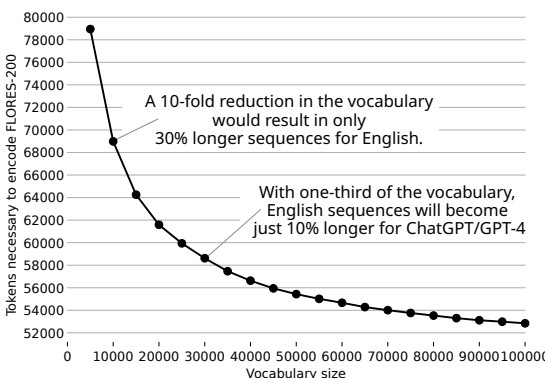

Figure 3: How much longer will English language tokenization be if we dedicate a fraction of the `cl100k_base` vocabulary to other languages? This plot shows how many tokens will be necessary to encode the English language corpus of FLORES-200 for different subsets of the `cl100k_base` vocabulary.

the input can easily be applied to existing systems without the need to retrain the LLM itself. However, as the tokenizer for the language model is unchanged, this approach would still suffer from latency and inability to process long contexts. Therefore, to ensure similar processing times and long context capabilities across languages, the language model has to be trained with a multilingually fair tokenizer.

**The tokenization needs to support all Unicode codepoints.** Amongst all tokenizers we examine in this paper, the ones which encode all FLORES-200 languages all have one thing in common: they build their tokenization on top of Unicode representation, allowing them them to represent all characters. Therefore, a multilingually fair tokenizer should also start from a Unicode (or equivalent) encoding. Considering that subword tokenization is necessary, building the vocabulary from UTF-8 would likely result in a smaller dictionary than building it on top of UTF-32. Hence, UTF-8 is likely the more appropriate choice.

**Building a multilingually fair parallel corpus.** Building and evaluating multilingually fair tokenizers requires attention to the parallel corpus used. One must ensure a balanced representation of topics, otherwise, the resulting tokenizer might end up being multilingually fair only for a subset of topics. The presence of named entities must also be balanced. For example, in FLORES-200, there are many English-centric names and institutions, which might skew the results in favour of English. Additionally, the same sentence can have different translations with varying tokenization lengths. To account for this, a diversity of translations could ensure tokenization fairness across languages. These limitations also hold for the results in this paper. Hence, developing a well-curated and diverse parallel corpus is crucial for the development and evaluation of a multilingually fair tokenizer.

**Building a multilingually fair tokenizer from monolinugal tokenizers.** As discussed in Section 4, byte-level, character-level and word-level tokenizers cannot achieve tokenization parity and subword tokenization is needed. However, simply training a subword tokenizer on a balanced dataset is also not sufficient as languages can share tokens. For example, "hotel" is written the same way in English, Spanish, Italian, Portuguese, Dutch, Danish, Hungarian, Polish, etc. Hence, languages from more numerous language families will also witness shorter tokenization lengths while more isolated languages and scripts, e.g. Korean, would see larger language premiums: "hotel" in Korean is "호텔" and no other language has the same spelling as no other language uses the Korean script.

To address this issue, we suggest a two-stage process towards building a multilingually fair tokenizer. First, train individual monolingual tokenizers for all target languages. Then, merge them while maintaining parity. The merging can be done by starting with the 256 tokens corresponding to each value a byte can take and then repeatedly adding the most frequently used token for the language with the highest premium.

While a multilingually fair tokenizer would lead to more tokens being needed for the dominant language, this additional cost would likely be much smaller than the benefit for the rest of the languages. The vocabulary size has diminishing returns: the additional tokens correspond to increasingly rare (parts of) words. For example, with only a third of the vocab-

ulary, English sequences will become just 10% longer for ChatGPT/GPT-4 (see Figure 3). Therefore, by removing rarely used tokens of the dominant language and replacing them with frequently used tokens in other languages, we would likely see an overall net benefit.

## 7  Related Works

**Fairness and bias in language models.** The rapid increase in the size of language models has raised concerns regarding their biases and unfairness (Bender et al., 2021). For example, Bolukbasi et al. (2016), May et al. (2019) and Nadeem et al. (2021) showed that stereotypes and biases exist in language models, while Magee et al. (2021) identified the presence of intersectional biases which may be resistant to debiasing techniques. Language models were also shown to rely on social biases in question answering (Parrish et al., 2022). Another challenge is the generation of toxic content which can occur even without prompting (Gehman et al., 2020). Interestingly, Gururangan et al. (2022) point out that datasets consider one type of English as a higher quality depending on the location of the writer rather than on factuality or literary acclaim. Moreover, Ramesh et al. (2023) and Levy et al. (2023) highlighted the need to consider fairness issues of languages other than English, as they may have distinct sources of bias and solutions for English may not be applicable.

**Multilingual performance.** One approach towards similar multilingual performance is to frame languages as entities as recently proposed by Choudhury and Deshpande (2021). Another method is to separately train vocabularies for different language clusters to balance cross-lingual and language-specific tokens (Chung et al., 2020). Still, multilingual models struggle to deliver on the promises of deep transfer learning for lower-resourced languages (Virtanen et al., 2019) and perform differently depending on the script and resource level of the language (Bang et al., 2023). Ahuja et al. (2023) found that generative models perform better on higher-resource languages and languages that use the Latin script, possibly due to the context length restrictions for some languages. Zhang et al. (2022) show that a balanced tokenizer corpus results in better translation performance. Separately, Hofmann et al. (2021, 2022) show that the BPE results in suboptimal token choices even for English and demonstrate that addressing this issue boosts performance. Similarly, Rajab (2022) and Oladipo et al. (2022) discuss how tokenization affects performance for African languages.

**Measuring tokenization lengths.** Zhang et al. (2022) suggested using the ratio of the average sentence length in tokens to the length in characters as a measure of closeness to the character level. However, this method may not be suitable for comparing languages due to differences in sentence length across languages. On the other hand, Ács (2019) and Scao et al. (2022) measure the number of tokens created per word, but this method may not be effective for comparing languages due to differences in semantic content per word and the lack of word delineation in some languages. Rust et al. (2021) show that mBERT (Devlin et al., 2019) breaks down English words the least, in line with our findings of English receiving special treatment. However, to the best of our knowledge, we are the first to leverage a parallel corpus to compare tokenization lengths across languages.

## 8  Conclusion

This paper highlights the significant disparities in tokenization across different languages which can lead to unequal treatment and disadvantages for certain language communities. The findings reveal that even tokenizers explicitly trained for multilingual support exhibit tokenization lengths that vary by up to a factor of 13. Furthermore, character-level and byte-level models also demonstrate encoding length discrepancies that are more than 4 times longer. These disparities have important real-world implications including increased costs for accessing commercial language services, longer processing times and limitations on the amount of contextual information provided to language models. To address these issues, we propose the development of multilingually fair tokenizers for future language models emphasizing the importance of ensuring comparable performance and accessibility across supported languages. By achieving tokenization parity, we can mitigate inequalities and promote fair access to language technologies across diverse linguistic communities.

## Acknowledgements

We would like to thank Puyu Wang, Francisco Eiras, Ambre Bertrand and Carmen Scheidemann for their linguistic advice. Janet Pierrehumbert introduced us to many relevant prior works. We also extend special gratitude to Shinnosuke Takamichi and Hiroshi Saruwatari for open-sourcing the CPJD corpus for this project. Finally, we thank the reviewers; their feedback greatly improved this manuscript.

AB has received funding from the Amazon Research Awards. This work is supported by a UKRI grant Turing AI Fellowship (EP/W002981/1) and the EPSRC Centre for Doctoral Training in Autonomous Intelligent Machines and Systems (EP/S024050/1). We also thank the Royal Academy of Engineering and FiveAI.

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

# A  Background on Tokenization

To enable automatic processing of language, it must first be represented in a suitable form. The current practice is to use *tokenization* which is the process of turning natural language into sequences of *tokens* coming from a finite and pre-determined set called *vocabulary* (Webster and Kit, 1992). Each token is typically associated with an integer value. Language models process such sequences of integers, rather than sequences of characters or words. In this section, we offer a brief overview of the contemporary tokenization methods. For further details, we recommend the comprehensive survey by Mielke et al. (2021).

**Word tokenization.**  The simplest tokenization method is splitting at white spaces, where each word is assigned its own token (Bengio et al., 2000). This approach, however, requires that all possible words are in the vocabulary which is not possible in practice. Therefore word tokenization often fails to handle cases like "won't", words spelled with accented characters like "naïve" or "açaí", speling mistakes and named entities like "Cottonshopeburnfoot" (Sun et al., 2020). This makes it unsuitable for representing *open vocabularies*, where the words encountered are not limited to a predetermined set. Furthermore, languages that do not use spaces to separate words, such as Chinese, Japanese and Burmese, pose additional challenges for this approach (Shao et al., 2018).

**Subword tokenization.**  Hence, most current models use *subword tokenization*, where complex words are broken down into multiple tokens. Subword tokenization can efficiently handle complex terms by breaking them down into parts, *e.g.*, "Cottonshopeburnfoot" → "Cotton"+"shop"+"e"+"burn"+"foot". This approach can represent novel words, including misspelled ones, in an open vocabulary setting.

Subword vocabularies are usually data-based approaches which use large corpora to learn which subword sequences occur frequently in practice. Schuster and Nakajima (2012) introduced one of the first subword tokenizers, WordPiece, as a way to handle Japanese and Korean. Sennrich et al. (2016) proposed using Byte-Pair Encoding (BPE) (Gage, 1994) for learning subwords by merging the most frequently occurring pairs. BPE has since been widely used for most of the popular tokenizers. Kudo (2018) proposed an alternative approach via gradually pruning a large vocabulary. It removes tokens that are less likely to improve the performance of a simple unigram language model. Both methods rely on pre-tokenization (splitting on whitespaces, when available), which is not an invertible process. SentencePiece (Kudo and Richardson, 2018) addresses this de-tokenization ambiguity by treating whitespace as a special symbol, including it in the vocabulary, and supports both methods. SentencePiece with BPE is by far the most popular tokenization method for the models considered in this paper.

**Unicode support.**  Even if subword tokenization ensures that individual characters are in the vocabulary, this still leaves the question of which characters are to be included. Simple solution is to take the ASCII characters. However, this means that words in other scripts or accented letters will fall out of it. A common workaround is to represent strings outside the vocabulary as a special UNK token. However, if there are too many UNK tokens in an input, the performance of the model tends to deteriorate (Pfeiffer et al., 2021). Therefore, it is desirable that the number of UNK tokens in the input is kept as low as possible. A simple and commonly used solution is to base the vocabulary building on Unicode.

Unicode is a computing industry standard for representing text characters (The Unicode Consortium, 2022). Unicode supports virtually all languages (including many ancient ones, emojis and special characters) by assigning every grapheme, modifier, punctuation mark, control character or formatting character one of 1,114,112 integer *codepoints*. The codepoints can be represented in binary as the variable-width encoding UTF-8, which encodes every codepoint with one to four bytes, or the fixed-width UTF-32 which encodes all codepoints with four bytes (see Figure 4).

UTF-8 can therefore represent any string in any language as a string of bytes. As each byte can take only one out of 256 values, 256 tokens can be sufficient to encode all texts. In practice this is usually combined with the BPE tokenizer. At first, the corpus is en-

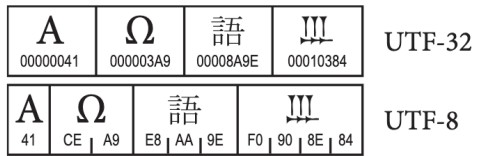

Figure 4: Comparison of variable width Unicode encoding (UTF-8) and fixed width encoding (UTF-32). Image adapted from (The Unicode Consortium, 2022).

coded as UTF-8 bytes and then BPE is ran on top of it. As most characters occur frequently, BPE would assign them a dedicated token. If the model encounters a character that didn't exist in the training corpus (*e.g.*, the medium skin tone waving hand 👋), it can still represent it byte-by-byte (F0+9F+91+8B for the waving hand and F0+9F+8F+BD for the skin tone modifier). This allows the vocabulary to efficiently represent frequently occurring words and rare characters. For example, the sentence "I love açaí" could be tokenized as "I "+"love "+"a"+C3+A7+"a"+C3+AD.

**Byte-level and character-level tokenization.** If we can represent any input with just 256 characters, then why bother with subword tokens? A key consideration is sequence length. This is since transformers (Vaswani et al., 2017), the currently predominant deep learning architecture for language models, have attention layers with a quadratic complexity in the input length. Hence, as the number of characters is much longer than the sub-word tokenization, working on the character level has been traditionally considered computationally inefficient. However, Chung et al. (2016), Lee et al. (2017), Gao et al. (2020), Clark et al. (2022) and Xue et al. (2022) proposed various architectures working around this issue and operating directly on characters or UTF-8 bytes.

# B  Parity for Linguistic Varieties

A language can vary according to factors such as geography, history, social class and culture. As a result, different dialects, pidgin and creole language variations emerge, each with its own distinct set of grammar, vocabulary and pronunciation rules.[9] Unequal treatment of certain dialects or languages can lead to social and economic disadvantages for those who speak them. Therefore, it is important to also study the tokenization differences between the "standard" language and its varieties.[10] Unfortunately, parallel corpora for dialects, pidgin and creole language variations are far and few in between. In this section, however, we show results on regional Swiss German varieties, Arabic and Japanese dialects, as well as Haitian and Mauritian creoles.

**Swiss German dialects.** Swiss German is a dialect continuum which significantly differs from the formal High German. German-speaking Switzerland is diglossic:[11] High German is used alongside regional dialects (Hogg et al., 1984). In contrast to other dialects, the use of Swiss dialects is increasing (Sieber and Sitta, 1987) especially online (Lüdi, 2007). Swiss German dialects are often considered unintelligible to High German speakers and sometimes even speakers of different dialects may find difficulty understanding each other (Russ, 1990). Therefore, ensuring that German-targeting NLP applications can process Swiss German dialects is important.

To this end, we compare the tokenization parity relative to High German of GottBERT (Scheible et al., 2020) on the regional dialects of Aargau, Bern, Basel, Graubünden, Luzern,

---

[9]While no standard definitions exist, dialects are usually considered to be regional variations of a language, whereas pidgin and creole languages are contact languages that emerge from the interaction of speakers of different languages (Muysken and Smith, 1994).

[10]We refer to the language that the datasets label as "standard", "official" or "dominant" without necessarily endorsing this designation.

[11]Diglossia is the situation of two dialects or languages being used by a single language community (Kaye, 2001).

Table 6: GottBERT tokenizer premiums on the SwissDial dataset for **Swiss German dialects**. The premium is computed with respect to High German.

| Region | GottBERT parity |
|---|---|
| High German | 1.00 |
| Zürich | 1.38 |
| St. Gallen | 1.40 |
| Basel | 1.41 |
| Graubünden | 1.44 |
| Luzern | 1.52 |
| Aargau | 1.53 |
| Wallis | 1.58 |
| Bern | 1.59 |

Table 7: ArabicBERT tokenizer premiums on the MADAR dataset for **Arabic dialects**. The premium is computed relative to Standard Arabic.

| City | ArabicBERT | City | ArabicBERT |
|---|---|---|---|
| Jeddah | 0.91 | Sanaa | 1.01 |
| Doha | 0.92 | Beirut | 1.02 |
| Riyadh | 0.92 | Benghazi | 1.02 |
| Muscat | 0.94 | Cairo | 1.03 |
| Basra | 0.95 | Sfax | 1.03 |
| Salt | 0.95 | Tripoli | 1.05 |
| Baghdad | 0.96 | Aswan | 1.06 |
| Damascus | 0.97 | Alexandria | 1.06 |
| Aleppo | 0.97 | Tunis | 1.06 |
| Jerusalem | 0.97 | Algiers | 1.07 |
| Khartoum | 0.98 | Mosul | 1.10 |
| Amman | 0.99 | Fes | 1.11 |
| Std. Arabic | 1.00 | Rabat | 1.17 |

St. Gallen, Wallis and Zürich. We use SwissDial, a parallel multidialectal corpus, as the basis of comparison (Dogan-Schönberger et al., 2021). It is worth noting, that the dialect of each city and its corresponding region may differ significantly. Therefore there might be large variations within regions as well.

The results in Table 6 show a disparity between the tokenization lengths for High German and the Swiss dialects with a premium ranging from 1.38 for the Zürich dialect, or *Züritüütsch*, to 1.59 for the Bernese *Bärndütsch*. In fact, English has a lower premium than any Swiss dialect (1.35 on FLORES-200, Table 2) and the premium for Bernese German is close to the linguistically further Swedish (1.64) and Norwegian Bokmål (1.65). The following example from SwissDial shows how the sentence "Like he's waiting for something" has almost twice as long tokenization in Bernese German compared to High German:

| 963 | 15628 | 63 | 18 | 145 | 4 |
|---|---|---|---|---|---|
| Als | warte | er | auf | etwas | . |

| 1134 | 8808 | 226 | 751 | 2912 | 13621 | 288 | 361 | 67 | 11769 | 4 |
|---|---|---|---|---|---|---|---|---|---|---|
| Aus | wür | der | | uf | ö | p | is | war | tä | . |

The fact that the GottBERT tokenizer results in better parity for English, Swedish and Norwegian Bokmål than for Swiss German dialects highlights that it does not likely pick out stable linguistic constructs.

**Arabic dialects.** Similarly to Swiss German, Arabic is usually spoken in diglossic speech communities, where Modern Standard Arabic is spoken alongside at least one prestigious vernacular particular to the country or region (Bassiouney, 2009). As both Standard Arabic

Table 8: BERT Japanese tokenizer premiums on the CPJD dataset for **Japanese dialects**. The premium is computed with respect to Standard Japanese. The CPJD dataset consists of two parallel corpora with the dialects split across the two. Hence, we have also indicated the corpus for each dialect. Nara-ben has two entries as the dataset has transcriptions for two separate speakers. The suffix "-ben" (弁) means "speech" or "dialect".

| Dialect | Corpus | Parity | Dialect | Corpus | Parity |
|---|---|---|---|---|---|
| Akita-ben | 2 | 1.09 | Miyazaki-ben | 1 | 1.05 |
| Awa-ben | 2 | 1.09 | Morokata-ben | 1 | 1.15 |
| Fukui-ben | 2 | 1.04 | Nara-ben | 2 | 1.09 |
| Fukuoka-ben | 1 | 1.03 | Nara-ben | 2 | 1.03 |
| Hiroshima-ben | 1 | 1.02 | Okayama-ben | 1 | 1.15 |
| Hokkaido-ben | 2 | 1.06 | Oosaka-ben | 2 | 1.03 |
| Iwaki-ben | 2 | 1.08 | Saitama-ben | 1 | 1.01 |
| Iyo-ben | 1 | 1.05 | Tosa-ben | 1 | 1.03 |
| Izumo-ben | 1 | 1.10 | Toshu-ben | 1 | 1.06 |
| Kanazawa-ben | 2 | 1.11 | Tsugaru-ben | 1 | 1.09 |
| Kyokotoba | 2 | 1.07 | | | |

and its dialects are commonly used in written communication, it is vital that tokenizers handle them equally well.

To assess the performance of Arabic tokenizers, we compare the tokenization lengths of ArabicBERT (Safaya et al., 2020) across 25 Arabic dialects. To this end, we use the MADAR parallel corpus of Arabic dialects (Bouamor et al., 2018).

Table 7 shows the premiums relative to Standard Modern Arabic. The premium varies from 0.91 for the Jeddah dialect to 1.17 for the Rabat dialect. This is significantly lower than the premium for English (1.83 on FLORES-200 Table 2). The range is also much smaller than for the Swiss German dialects and approximately half of the considered dialects have a lower premium than Standard Modern Arabic. Therefore, one could say that the tokenizer of ArabicBERT achieves tokenization parity for these 25 Arabic vernaculars. This is likely because the corpus and vocabulary set on which ArabicBERT was trained contained dialectical Arabic. It is also possible that Arabic dialects are closer to Modern Standard Arabic and more mutually intelligible than Swiss German dialects are to High German (Čéplö et al., 2016; Trentman and Shiri, 2020). Still, this difference between the parity for Swiss and Arabic dialects indicates that including a broader set of vernaculars and dialects in the corpus results in improved tokenization parity.

**Japanese dialects.** Japanese also has a number of regional dialects (Hattori, 1973). We compare the tokenization parity of BERT Japanese (Tohoku NLP Group, 2019) across them. We employ the CPJD dataset by Takamichi and Saruwatari (2018) which contains transcriptions of the voice recordings of 250 sentences across 20 dialects.

The results in Table 8 show that the premium compared to Standard Japanese (Tokyo dialect) ranges from 1.01 (for Saitama prefecture, neighbouring Tokyo) to 1.15 (for Morokata-ben and Okayama-ben). These all are significantly lower than the premium for English (1.49, as shown in Table 2). Therefore, similarly to ArabicBERT, this is an example of the tokenizer being relatively well-aligned with the dialects. This is likely because Japanese dialects are more closely related (and intelligible (Yamagiwa, 1967) to Standard Japanese speakers) than the Swiss dialects are to High German speakers.

**Mauritian and Haitian Creoles.** While creoles often have some similarities with a high-resource language (usually English or French), the differences are significant to necessitate special attention to their support (Lent et al., 2021, 2022). This is especially critical for emergency services and disaster management (Munro, 2010).

Mauritian Creole is based on French as well as the languages of slaves imported from Madagascar and East Africa. As the British gained control of Mauritius, they brought indentured labourers from India who further had an effect on the formation of the modern Mauritian

Creole (Seuren, 1995). Similarly, Haitian Creole (*Kreyòl*) emerged from the interaction of French and the various Niger-Congo languages spoken by the Africans brought as slaves (DeGraff, 2007).

Considering that both languages have their basis in French, one would expect that tokenizers targeting French would have low tokenization parities for Mauritian and Haitian Creoles. However, taking the tokenizer of CamemBERT (Martin et al., 2020), the premium for Mauritian Creole is 1.20 using the MorisienMT parallel corpus (Dabre and Sukhoo, 2022). The premium for Haitian Creole is 1.64 when using the QEDv2 corpus (Tiedemann, 2012; Abdelali et al., 2014). Haitian Creole is also represented in the FLORES-200 dataset where the premium relative to French is 1.58. This is significantly larger than linguistically further languages such as English (1.20), Pangasinan (1.49) and Nigerian Fulfulde (1.54). Therefore, CamemBERT is not well-placed to tokenize French-related creoles despite the model being trained for French.

## C Extended Tables of Tokenization Premiums

In addition to the models presented in the main text, these extended tables also include LLAMA (Touvron et al., 2023), MBart50 (Liu et al., 2020; Tang et al., 2020), SeamlessM4T (Barrault et al., 2023) and Qwen-VL (Bai et al., 2023).

| Language | LLAMA | GPT-2 | r50k_base | p50k_base | p50k_edit | cl100k_base | RoBERTa | GottBERT | CamemBERT | PhoBERT | RoCBert | XLM-RoBERTa | M2M100 |
|---|---|---|---|---|---|---|---|---|---|---|---|---|---|
| Acehnese (Arabic script) | 4.00 | 4.78 | 4.78 | 4.78 | 4.78 | 3.78 | 4.78 | 4.95 | — | — | — | 1.94 | 1.89 |
| Acehnese (Latin script) | 1.89 | 2.16 | 2.16 | 2.16 | 2.16 | 1.98 | 2.16 | 1.56 | 1.55 | 1.37 | 1.10 | 1.57 | 1.47 |
| Mesopotamian Arabic | 3.34 | 4.27 | 4.27 | 4.27 | 4.27 | 2.99 | 4.27 | 5.10 | — | — | — | 1.16 | 1.27 |
| Ta'izzi-Adeni Arabic | 3.38 | 4.34 | 4.34 | 4.34 | 4.34 | 3.01 | 4.34 | 5.16 | — | — | — | 1.17 | 1.28 |
| Tunisian Arabic | 3.31 | 4.20 | 4.20 | 4.20 | 4.20 | 2.93 | 4.20 | 5.03 | — | — | — | 1.20 | 1.29 |
| Afrikaans | 1.55 | 1.94 | 1.94 | 1.94 | 1.94 | 1.69 | 1.94 | 1.25 | 1.38 | 1.26 | 1.06 | 1.20 | 1.22 |
| South Levantine Arabic | 3.20 | 4.02 | 4.02 | 4.02 | 4.02 | 2.84 | 4.02 | 4.84 | — | — | — | 1.12 | 1.22 |
| Akan | 2.20 | 2.80 | 2.80 | 2.80 | 2.80 | 2.68 | 2.80 | 1.90 | 1.64 | 1.45 | — | 1.98 | 1.83 |
| Tosk Albanian | 2.26 | 2.65 | 2.65 | 2.65 | 2.65 | 2.25 | 2.65 | 1.77 | 1.82 | 1.69 | 1.12 | 1.32 | 1.36 |
| Amharic | 7.32 | 7.79 | 7.79 | 7.79 | 7.79 | 7.68 | 7.79 | 5.19 | — | — | — | 1.34 | 1.42 |
| North Levantine Arabic | 3.19 | 4.04 | 4.04 | 4.04 | 4.04 | 2.83 | 4.04 | 4.83 | — | — | — | 1.15 | 1.24 |
| Standard Arabic | 3.42 | 4.40 | 4.40 | 4.40 | 4.40 | 3.04 | 4.40 | 5.21 | — | — | — | 1.18 | 1.29 |
| Standard Arabic (Romanized) | 2.31 | 2.51 | 2.51 | 2.51 | 2.51 | 2.45 | 2.51 | 1.76 | 1.72 | 1.55 | 1.19 | 1.94 | 1.83 |
| Najdi Arabic | 3.43 | 4.41 | 4.41 | 4.41 | 4.41 | 3.04 | 4.41 | 5.22 | — | — | — | 1.18 | 1.30 |
| Moroccan Arabic | 3.35 | 4.21 | 4.21 | 4.21 | 4.21 | 2.96 | 4.21 | 5.08 | — | — | — | 1.25 | 1.33 |
| Egyptian Arabic | 3.36 | 4.23 | 4.23 | 4.23 | 4.23 | 2.96 | 4.23 | 5.10 | — | — | — | 1.17 | 1.27 |
| Assamese | 6.14 | 9.79 | 9.79 | 9.78 | 9.78 | 6.20 | 9.79 | 8.32 | — | — | — | 1.90 | 2.24 |
| Asturian | 1.48 | 1.89 | 1.89 | 1.89 | 1.89 | 1.58 | 1.89 | 1.33 | 1.31 | 1.24 | 1.04 | 1.27 | 1.15 |
| Awadhi | 4.53 | 7.19 | 7.19 | 7.19 | 7.19 | 4.78 | 7.19 | 8.19 | — | — | — | 1.37 | 1.47 |
| Central Aymara | 2.03 | 2.32 | 2.32 | 2.32 | 2.32 | 2.17 | 2.32 | 1.62 | 1.62 | 1.47 | 1.09 | 1.70 | 1.64 |
| South Azerbaijani | 3.76 | 5.16 | 5.16 | 5.16 | 5.16 | 3.34 | 5.16 | 5.32 | — | — | — | 1.43 | 1.50 |
| North Azerbaijani | 2.61 | 3.47 | 3.47 | 3.47 | 3.47 | 2.64 | 3.47 | 2.31 | — | 1.90 | — | 1.15 | 1.26 |
| Bashkir | 2.91 | 6.01 | 6.01 | 6.01 | 6.01 | 4.28 | 6.01 | 3.97 | — | — | — | 2.06 | 1.23 |
| Bambara | 1.99 | 2.66 | 2.66 | 2.66 | 2.66 | 2.57 | 2.66 | 1.84 | 1.54 | 1.40 | — | 1.82 | 1.72 |
| Balinese | 1.77 | 1.97 | 1.97 | 1.97 | 1.97 | 1.80 | 1.97 | 1.39 | 1.43 | 1.28 | 1.14 | 1.32 | 1.29 |
| Belarusian | 2.38 | 6.56 | 6.56 | 6.56 | 6.56 | 3.55 | 6.56 | 4.17 | — | 2.88 | — | 1.46 | 1.56 |
| Bemba | 2.15 | 2.46 | 2.46 | 2.46 | 2.46 | 2.23 | 2.46 | 1.69 | 1.68 | 1.53 | 1.26 | 1.76 | 1.67 |
| Bengali | 5.38 | 9.65 | 9.65 | 9.65 | 9.65 | 5.84 | 9.65 | 8.54 | — | — | — | 1.38 | 1.55 |
| Bhojpuri | 4.52 | 7.18 | 7.18 | 7.18 | 7.18 | 4.69 | 7.18 | 8.08 | — | — | — | 1.47 | 1.54 |
| Banjar (Arabic script) | 4.22 | 5.03 | 5.03 | 5.03 | 5.03 | 3.80 | 5.03 | 5.53 | — | — | — | 1.92 | 1.93 |
| Banjar (Latin script) | 1.75 | 1.98 | 1.98 | 1.98 | 1.98 | 1.71 | 1.98 | 1.38 | 1.35 | 1.21 | 1.08 | 1.21 | 1.16 |
| Standard Tibetan | 6.67 | 14.93 | 14.93 | 14.93 | 14.93 | 11.27 | 14.93 | 10.87 | — | — | — | — | — |
| Bosnian | 1.69 | 2.19 | 2.19 | 2.19 | 2.19 | 1.87 | 2.19 | 1.47 | 1.46 | 1.35 | 1.02 | 1.12 | 1.17 |
| Buginese | 1.87 | 2.20 | 2.20 | 2.20 | 2.20 | 1.98 | 2.20 | 1.49 | 1.45 | 1.35 | 1.10 | 1.51 | 1.49 |
| Bulgarian | 1.78 | 5.51 | 5.51 | 5.51 | 5.51 | 2.64 | 5.51 | 3.51 | — | 2.57 | — | 1.16 | 1.23 |
| Catalan | 1.51 | 1.92 | 1.92 | 1.92 | 1.92 | 1.71 | 1.92 | 1.40 | 1.33 | 1.31 | 1.10 | 1.26 | 1.26 |
| Cebuano | 1.96 | 2.24 | 2.24 | 2.24 | 2.24 | 1.93 | 2.24 | 1.57 | 1.59 | 1.41 | 1.20 | 1.52 | 1.38 |
| Czech | 1.69 | 2.62 | 2.62 | 2.62 | 2.62 | 2.11 | 2.62 | 1.73 | — | 1.48 | 0.99 | 1.17 | 1.23 |
| Chokwe | 1.91 | 2.16 | 2.16 | 2.16 | 2.16 | 1.98 | 2.16 | 1.51 | 1.49 | 1.32 | 1.10 | 1.55 | 1.47 |
| Central Kurdish | 4.43 | 6.49 | 6.49 | 6.49 | 6.49 | 4.80 | 6.49 | 5.82 | — | — | — | 2.30 | 2.48 |
| Crimean Tatar | 2.13 | 2.49 | 2.49 | 2.49 | 2.49 | 2.12 | 2.49 | 1.67 | 1.68 | 1.54 | — | 1.38 | 1.37 |
| Welsh | 2.09 | 2.34 | 2.34 | 2.34 | 2.34 | 2.12 | 2.34 | 1.66 | 1.68 | 1.53 | 1.06 | 1.43 | 1.44 |
| Danish | 1.54 | 1.90 | 1.90 | 1.90 | 1.90 | 1.62 | 1.90 | 1.26 | 1.39 | 1.29 | 1.04 | 1.09 | 1.12 |
| German | 1.41 | 2.14 | 2.14 | 2.14 | 2.14 | 1.58 | 2.14 | 0.74 | 1.55 | 1.40 | 1.20 | 1.17 | 1.24 |
| Southwestern Dinka | 1.88 | 2.48 | 2.48 | 2.48 | 2.48 | 2.25 | 2.48 | 1.60 | 1.43 | 1.32 | 0.75 | 1.68 | 1.55 |
| Dyula | 1.88 | 2.20 | 2.20 | 2.20 | 2.20 | 2.05 | 2.20 | 1.54 | 1.43 | 1.30 | 0.98 | 1.65 | 1.53 |
| Dzongkha | 7.42 | 16.36 | 16.36 | 16.36 | 16.36 | 12.33 | 16.36 | 11.95 | — | — | — | — | — |
| Greek | 4.99 | 6.54 | 6.54 | 6.54 | 6.54 | 5.15 | 6.54 | 4.99 | — | 3.11 | 1.15 | 1.45 | 1.58 |
| English | 1.00 | 1.00 | 1.00 | 1.00 | 1.00 | 1.00 | 1.00 | 1.00 | 1.00 | 1.00 | 1.00 | 1.00 | 1.00 |
| Esperanto | 1.67 | 2.03 | 2.03 | 2.03 | 2.03 | 1.87 | 2.03 | 1.37 | 1.35 | 1.26 | 1.01 | 1.20 | 1.38 |
| Estonian | 1.76 | 2.11 | 2.11 | 2.11 | 2.11 | 1.87 | 2.11 | 1.39 | 1.42 | 1.33 | 1.03 | 1.12 | 1.20 |
| Basque | 1.79 | 2.10 | 2.10 | 2.10 | 2.10 | 1.88 | 2.10 | 1.39 | 1.44 | 1.33 | 1.11 | 1.16 | 1.23 |
| Ewe | 2.28 | 2.90 | 2.90 | 2.90 | 2.90 | 2.75 | 2.90 | 1.97 | 1.69 | 1.46 | — | 2.01 | 1.86 |
| Faroese | 1.92 | 2.38 | 2.38 | 2.38 | 2.38 | 2.07 | 2.38 | 1.66 | 1.64 | 1.46 | — | 1.44 | 1.41 |
| Fijian | 2.02 | 2.30 | 2.30 | 2.30 | 2.30 | 2.15 | 2.30 | 1.67 | 1.52 | 1.39 | 1.13 | 1.72 | 1.62 |
| Finnish | 1.91 | 2.28 | 2.28 | 2.28 | 2.28 | 1.99 | 2.28 | 1.46 | 1.56 | 1.47 | 1.13 | 1.14 | 1.23 |
| Fon | 2.83 | 4.08 | 4.08 | 4.08 | 4.08 | 3.67 | 4.08 | 2.75 | — | — | — | 2.51 | 2.31 |
| French | 1.47 | 2.00 | 2.00 | 2.00 | 2.00 | 1.60 | 2.00 | 1.47 | 0.84 | 1.38 | 1.20 | 1.30 | 1.33 |
| Friulian | 1.70 | 2.07 | 2.07 | 2.07 | 2.07 | 1.85 | 2.07 | 1.47 | 1.38 | 1.33 | 1.07 | 1.56 | 1.47 |
| Nigerian Fulfulde | 1.72 | 1.99 | 1.99 | 1.99 | 1.99 | 1.85 | 1.99 | 1.37 | 1.29 | 1.16 | 0.86 | 1.46 | 1.27 |
| West Central Oromo | 2.22 | 2.53 | 2.53 | 2.53 | 2.53 | 2.32 | 2.53 | 1.72 | 1.73 | 1.61 | 1.24 | 1.78 | 1.49 |
| Scottish Gaelic | 2.33 | 2.70 | 2.70 | 2.70 | 2.70 | 2.42 | 2.70 | 1.86 | 1.80 | 1.61 | 1.24 | 1.75 | 1.61 |
| Irish | 2.17 | 2.56 | 2.56 | 2.56 | 2.56 | 2.33 | 2.56 | 1.76 | 1.75 | 1.55 | 1.15 | 1.50 | 1.50 |
| Galician | 1.48 | 1.91 | 1.91 | 1.91 | 1.91 | 1.56 | 1.91 | 1.39 | 1.36 | 1.30 | 1.11 | 1.13 | 1.14 |
| Guarani | 1.99 | 2.46 | 2.46 | 2.46 | 2.46 | 2.17 | 2.46 | 1.68 | 1.55 | 1.45 | 1.05 | 1.72 | 1.63 |
| Gujarati | 9.98 | 12.27 | 12.27 | 12.27 | 12.27 | 7.69 | 12.27 | 8.17 | — | — | — | 1.42 | 1.58 |
| Haitian Creole | 1.58 | 1.90 | 1.90 | 1.90 | 1.90 | 1.74 | 1.90 | 1.35 | 1.32 | 1.15 | 0.89 | 1.39 | 1.16 |
| Hausa | 1.89 | 2.15 | 2.15 | 2.15 | 2.15 | 2.00 | 2.15 | 1.49 | 1.47 | 1.26 | 1.02 | 1.40 | 1.29 |

| Language | MBart50 | mT5 | FlanT5 | ByT5 | CANINE | BLOOM | ArabicBERT | MuRIL | UTF-32 | BERT Japanese | SeamlessM4T | NLLB | Qwen |
|---|---|---|---|---|---|---|---|---|---|---|---|---|---|
| Acehnese (Arabic script) | 1.94 | 1.79 | — | 1.51 | 0.85 | 2.65 | — | — | 0.85 | — | 1.89 | 1.89 | 2.66 |
| Acehnese (Latin script) | 1.57 | 1.44 | 2.55 | 1.09 | 1.07 | 1.74 | 1.44 | 2.02 | 1.07 | 1.41 | 1.24 | 1.24 | 1.95 |
| Mesopotamian Arabic | 1.16 | 1.28 | — | 1.56 | 0.86 | 1.15 | 0.55 | 1.93 | 0.86 | — | 1.37 | 1.37 | 1.63 |
| Ta'izzi-Adeni Arabic | 1.17 | 1.32 | — | 1.58 | 0.87 | 1.15 | 0.55 | 1.94 | 0.87 | — | 1.39 | 1.39 | 1.63 |
| Tunisian Arabic | 1.20 | 1.29 | — | 1.54 | 0.85 | 1.19 | 0.57 | 1.90 | 0.85 | — | 1.39 | 1.39 | 1.66 |
| Afrikaans | 1.20 | 1.20 | 2.15 | 1.07 | 1.06 | 1.69 | 1.33 | 1.84 | 1.06 | 1.27 | 1.22 | 1.22 | 1.67 |
| South Levantine Arabic | 1.12 | 1.24 | — | 1.49 | 0.83 | 1.12 | 0.55 | 1.82 | 0.83 | — | 1.31 | 1.31 | 1.55 |
| Akan | 1.98 | 1.82 | 2.96 | 1.10 | 1.00 | 2.05 | — | — | 1.00 | 1.45 | 1.40 | 1.40 | 2.28 |
| Tosk Albanian | 1.32 | 1.48 | 3.09 | 1.20 | 1.12 | 2.17 | 1.46 | 2.52 | 1.12 | — | 1.35 | 1.35 | 2.23 |
| Amharic | 1.34 | 1.73 | — | 1.72 | 0.67 | 5.07 | — | — | 0.67 | — | 1.32 | 1.32 | 4.16 |
| North Levantine Arabic | 1.15 | 1.23 | — | 1.48 | 0.82 | 1.13 | 0.55 | 1.83 | 0.82 | — | 1.33 | 1.33 | 1.58 |
| Standard Arabic | 1.18 | 1.35 | — | 1.60 | 0.88 | 1.14 | 0.55 | 1.97 | 0.88 | — | 1.40 | 1.40 | 1.63 |
| Standard Arabic (Romanized) | 1.94 | 1.73 | 2.94 | 1.17 | 1.17 | 2.15 | 1.60 | 2.28 | 1.17 | 1.64 | 1.86 | 1.86 | 2.42 |
| Najdi Arabic | 1.18 | 1.35 | — | 1.60 | 0.88 | 1.15 | 0.55 | 1.97 | 0.88 | — | 1.40 | 1.40 | 1.63 |
| Moroccan Arabic | 1.25 | 1.29 | — | 1.56 | 0.86 | 1.26 | 0.63 | 1.91 | 0.86 | — | 1.39 | 1.39 | 1.70 |
| Egyptian Arabic | 1.17 | 1.28 | — | 1.56 | 0.86 | 1.16 | 0.57 | 1.89 | 0.86 | — | 1.36 | 1.36 | 1.64 |
| Assamese | 1.90 | 1.94 | — | 2.54 | 0.96 | 1.41 | — | 1.24 | 0.96 | — | 1.39 | 1.39 | 5.46 |
| Asturian | 1.27 | 1.28 | 2.07 | 1.07 | 1.03 | 1.31 | 1.24 | 1.81 | 1.03 | 1.26 | 1.17 | 1.17 | 1.56 |
| Awadhi | 1.37 | 1.62 | — | 2.50 | 0.98 | 1.43 | — | 1.29 | 0.98 | — | 1.22 | 1.22 | 4.36 |
| Central Aymara | 1.70 | 1.57 | 2.71 | 1.07 | 1.05 | 1.94 | 1.44 | 1.98 | 1.05 | 1.45 | 1.32 | 1.32 | 2.15 |
| South Azerbaijani | 1.43 | 1.42 | — | 1.63 | 0.89 | 1.81 | 1.11 | 1.72 | 0.89 | — | 1.37 | 1.37 | 2.62 |
| North Azerbaijani | 1.15 | 1.35 | — | 1.26 | 1.09 | 2.30 | 1.74 | — | 1.09 | — | 1.33 | 1.33 | 2.49 |
| Bashkir | 2.06 | 1.60 | — | 1.85 | 1.01 | 3.57 | — | — | 1.01 | — | 1.22 | 1.22 | 3.14 |
| Bambara | 1.82 | 1.65 | 2.70 | 1.04 | 0.96 | 1.89 | — | — | 0.96 | 1.34 | 1.27 | 1.27 | 2.14 |
| Balinese | 1.32 | 1.29 | 2.37 | 1.11 | 1.11 | 1.46 | 1.40 | 1.83 | 1.11 | 1.35 | 1.08 | 1.08 | 1.79 |
| Belarusian | 1.46 | 1.59 | — | 2.06 | 1.13 | 3.24 | 2.60 | — | 1.13 | — | 1.72 | 1.72 | 3.00 |
| Bemba | 1.76 | 1.57 | 3.01 | 1.23 | 1.23 | 1.92 | 1.65 | 2.17 | 1.23 | 1.64 | 1.39 | 1.39 | 2.20 |
| Bengali | 1.38 | 1.58 | — | 2.61 | 0.98 | 1.17 | — | 1.01 | 0.98 | — | 1.28 | 1.28 | 5.09 |
| Bhojpuri | 1.47 | 1.63 | — | 2.47 | 0.97 | 1.53 | — | 1.39 | 0.97 | — | 1.28 | 1.28 | 4.33 |
| Banjar (Arabic script) | 1.92 | 1.76 | — | 1.69 | 0.93 | 2.47 | 1.04 | — | 0.93 | — | 1.88 | 1.88 | 2.63 |
| Banjar (Latin script) | 1.21 | 1.16 | 2.20 | 1.05 | 1.05 | 1.30 | 1.32 | 1.71 | 1.05 | 1.29 | 1.08 | 1.08 | 1.70 |
| Standard Tibetan | — | 3.68 | — | 3.31 | 1.13 | 6.66 | — | — | 1.13 | — | 1.44 | 1.44 | 7.33 |
| Bosnian | 1.12 | 1.33 | 2.48 | 1.03 | 1.01 | 1.84 | 1.39 | — | 1.01 | 1.30 | 1.19 | 1.19 | 1.86 |
| Buginese | 1.51 | 1.44 | 2.51 | 1.09 | 1.06 | 1.71 | 1.45 | 1.96 | 1.06 | 1.39 | 1.30 | 1.30 | 1.96 |
| Bulgarian | 1.16 | 1.28 | — | 1.89 | 1.04 | 2.49 | 2.35 | — | 1.04 | — | 1.31 | 1.31 | 2.20 |
| Catalan | 1.26 | 1.36 | 2.14 | 1.12 | 1.10 | 1.18 | 1.29 | 1.90 | 1.10 | 1.30 | 1.25 | 1.25 | 1.69 |
| Cebuano | 1.52 | 1.42 | 2.86 | 1.20 | 1.20 | 1.78 | 1.51 | 2.10 | 1.20 | 1.53 | 1.29 | 1.29 | 1.91 |
| Czech | 1.17 | 1.27 | 2.72 | 1.08 | 0.97 | 2.03 | 1.31 | — | 0.97 | — | 1.26 | 1.26 | 2.07 |
| Chokwe | 1.55 | 1.41 | 2.66 | 1.07 | 1.07 | 1.72 | 1.47 | 1.94 | 1.07 | 1.42 | 1.34 | 1.34 | 1.94 |
| Central Kurdish | 2.30 | 1.75 | — | 1.78 | 0.97 | 3.21 | 1.65 | — | 0.97 | — | 1.30 | 1.30 | 3.46 |
| Crimean Tatar | 1.38 | 1.32 | 2.80 | 1.13 | 1.03 | 2.07 | 1.45 | — | 1.03 | — | 1.25 | 1.25 | 1.95 |
| Welsh | 1.43 | 1.70 | 3.12 | 1.07 | 1.07 | 2.09 | 1.55 | 2.32 | 1.07 | 1.47 | 1.38 | 1.38 | 2.09 |
| Danish | 1.09 | 1.14 | 2.26 | 1.05 | 1.03 | 1.67 | 1.28 | 1.83 | 1.03 | — | 1.11 | 1.11 | 1.61 |
| German | 1.17 | 1.19 | 1.37 | 1.18 | 1.17 | 1.68 | 1.44 | 2.02 | 1.17 | 1.37 | 1.29 | 1.29 | 1.55 |
| Southwestern Dinka | 1.68 | 1.58 | — | 0.96 | 0.86 | 1.82 | — | — | 0.86 | — | 1.25 | 1.25 | 2.01 |
| Dyula | 1.65 | 1.55 | 2.68 | 1.07 | 1.01 | 1.80 | 1.30 | 2.06 | 1.01 | 1.39 | 1.44 | 1.44 | 1.96 |
| Dzongkha | — | 4.24 | — | 3.64 | 1.25 | 7.36 | — | — | 1.25 | — | 1.48 | 1.48 | 8.19 |
| Greek | 1.45 | 1.65 | — | 2.17 | 1.20 | 3.81 | 2.70 | — | 1.20 | — | 1.65 | 1.65 | 4.95 |
| English | 1.00 | 1.00 | 1.00 | 1.00 | 1.00 | 1.00 | 1.00 | 1.00 | 1.00 | 1.00 | 1.00 | 1.00 | 1.00 |
| Esperanto | 1.20 | 1.19 | 2.19 | 1.02 | 1.00 | 1.65 | 1.24 | — | 1.00 | — | 1.23 | 1.23 | 1.80 |
| Estonian | 1.12 | 1.12 | 2.43 | 1.01 | 0.98 | 1.77 | 1.28 | 1.71 | 0.98 | — | 1.16 | 1.16 | 1.85 |
| Basque | 1.16 | 1.22 | 2.33 | 1.07 | 1.06 | 1.14 | 1.41 | 1.90 | 1.06 | 1.35 | 1.27 | 1.27 | 1.87 |
| Ewe | 2.01 | 1.82 | 2.85 | 1.07 | 0.97 | 2.11 | — | — | 0.97 | — | 1.27 | 1.27 | 2.36 |
| Faroese | 1.44 | 1.40 | 2.73 | 1.09 | 1.02 | 1.95 | 1.41 | — | 1.02 | — | 1.31 | 1.31 | 2.04 |
| Fijian | 1.72 | 1.59 | 3.02 | 1.17 | 1.17 | 1.99 | 1.65 | 2.01 | 1.17 | 1.53 | 1.32 | 1.32 | 2.13 |
| Finnish | 1.14 | 1.16 | 2.61 | 1.11 | 1.07 | 1.89 | 1.42 | 2.05 | 1.07 | 1.45 | 1.21 | 1.21 | 1.97 |
| Fon | 2.51 | 2.36 | — | 1.26 | 1.02 | 2.21 | — | — | 1.02 | — | 1.59 | 1.59 | 2.87 |
| French | 1.30 | 1.40 | 1.60 | 1.24 | 1.19 | 1.20 | 1.33 | 1.96 | 1.19 | 1.36 | 1.35 | 1.35 | 1.57 |
| Friulian | 1.56 | 1.52 | 2.30 | 1.13 | 1.10 | 1.70 | 1.28 | 1.94 | 1.10 | 1.29 | 1.37 | 1.37 | 1.83 |
| Nigerian Fulfulde | 1.46 | 1.32 | 2.14 | 0.96 | 0.93 | 1.66 | 1.16 | 1.54 | 0.93 | 1.21 | 1.24 | 1.24 | 1.75 |
| West Central Oromo | 1.78 | 1.69 | 3.16 | 1.20 | 1.19 | 2.19 | 1.63 | 2.17 | 1.19 | 1.63 | 1.42 | 1.42 | 2.29 |
| Scottish Gaelic | 1.75 | 1.85 | 3.24 | 1.28 | 1.24 | 2.25 | 1.57 | 2.27 | 1.24 | 1.49 | 1.56 | 1.56 | 2.38 |
| Irish | 1.50 | 1.67 | 3.14 | 1.23 | 1.16 | 2.15 | 1.45 | 2.46 | 1.16 | 1.51 | 1.42 | 1.42 | 2.28 |
| Galician | 1.13 | 1.31 | 2.18 | 1.13 | 1.11 | 1.27 | 1.30 | 1.91 | 1.11 | 1.32 | 1.16 | 1.16 | 1.54 |
| Guarani | 1.72 | 1.62 | 2.57 | 1.09 | 1.01 | 1.87 | 1.40 | 1.99 | 1.01 | — | 1.34 | 1.34 | 2.09 |
| Gujarati | 1.42 | 1.73 | — | 2.50 | 0.96 | 1.35 | — | 1.19 | 0.96 | — | 1.35 | 1.35 | 6.78 |
| Haitian Creole | 1.39 | 1.22 | 2.32 | 0.95 | 0.92 | 1.56 | 1.18 | 1.68 | 0.92 | 1.19 | 1.11 | 1.11 | 1.72 |
| Hausa | 1.40 | 1.37 | 2.61 | 1.08 | 1.07 | 1.78 | 1.34 | 1.78 | 1.07 | 1.35 | 1.18 | 1.18 | 1.95 |

| Language | LLAMA | GPT-2 | r50k_base | p50k_base | p50k_edit | cl100k_base | RoBERTa | GottBERT | CamemBERT | PhoBERT | RoCBert | XLM-RoBERTa | M2M100 |
|---|---|---|---|---|---|---|---|---|---|---|---|---|---|
| Hebrew | 3.29 | 4.39 | 4.39 | 4.39 | 4.39 | 3.66 | 4.39 | 4.52 | — | — | — | 1.12 | 1.22 |
| Hindi | 4.60 | 7.46 | 7.46 | 7.46 | 7.46 | 4.79 | 7.46 | 8.34 | — | — | — | 1.25 | 1.36 |
| Chhattisgarhi | 4.44 | 7.21 | 7.21 | 7.21 | 7.21 | 4.69 | 7.21 | 8.05 | — | — | — | 1.41 | 1.51 |
| Croatian | 1.67 | 2.15 | 2.15 | 2.15 | 2.15 | 1.85 | 2.15 | 1.46 | 1.43 | 1.33 | 1.00 | 1.10 | 1.15 |
| Hungarian | 1.79 | 2.66 | 2.66 | 2.66 | 2.66 | 2.15 | 2.66 | 1.79 | 1.78 | 1.57 | 1.09 | 1.18 | 1.28 |
| Armenian | 5.11 | 10.01 | 10.01 | 10.01 | 10.01 | 9.98 | 10.01 | 6.67 | — | — | — | 1.38 | 1.50 |
| Igbo | 2.32 | 3.42 | 3.42 | 3.42 | 3.42 | 2.44 | 3.42 | 2.33 | 1.77 | 1.48 | 0.99 | 2.12 | 1.47 |
| Ilocano | 2.01 | 2.26 | 2.26 | 2.26 | 2.26 | 2.05 | 2.26 | 1.59 | 1.61 | 1.41 | 1.21 | 1.61 | 1.33 |
| Indonesian | 1.76 | 1.98 | 1.98 | 1.98 | 1.98 | 1.55 | 1.98 | 1.37 | 1.40 | 1.25 | 1.12 | 0.94 | 0.98 |
| Icelandic | 1.98 | 2.43 | 2.43 | 2.43 | 2.43 | 2.15 | 2.43 | 1.72 | — | 1.50 | — | 1.23 | 1.29 |
| Italian | 1.46 | 2.01 | 2.01 | 2.01 | 2.01 | 1.64 | 2.01 | 1.43 | 1.36 | 1.33 | 1.19 | 1.19 | 1.25 |
| Javanese | 1.72 | 1.93 | 1.93 | 1.93 | 1.93 | 1.73 | 1.93 | 1.36 | 1.39 | 1.21 | 1.06 | 1.15 | 1.10 |
| Japanese | 2.24 | 3.00 | 3.00 | 3.00 | 3.00 | 2.30 | 3.00 | 3.23 | — | — | 0.52 | 1.11 | 1.20 |
| Kabyle | 2.00 | 2.50 | 2.50 | 2.50 | 2.50 | 2.47 | 2.50 | 1.74 | 1.59 | 1.43 | 0.90 | 1.84 | 1.71 |
| Jingpho | 2.27 | 2.65 | 2.65 | 2.65 | 2.65 | 2.35 | 2.65 | 1.89 | 1.78 | 1.54 | 1.20 | 1.94 | 1.78 |
| Kamba | 1.91 | 2.32 | 2.32 | 2.32 | 2.32 | 2.17 | 2.32 | 1.62 | 1.48 | 1.30 | 0.98 | 1.62 | 1.52 |
| Kannada | 10.83 | 13.69 | 13.69 | 13.68 | 13.68 | 8.90 | 13.69 | 9.27 | — | — | — | 1.36 | 1.53 |
| Kashmiri (Arabic script) | 4.43 | 6.19 | 6.19 | 6.19 | 6.19 | 4.62 | 6.19 | 5.63 | — | — | — | 1.93 | 1.93 |
| Kashmiri (Devanagari script) | 4.44 | 7.03 | 7.03 | 7.03 | 7.03 | 4.69 | 7.03 | 7.76 | — | — | — | 1.82 | 1.86 |
| Georgian | 4.87 | 13.85 | 13.85 | 13.85 | 13.85 | 9.85 | 13.85 | 9.22 | — | — | — | 1.34 | 1.56 |
| Kazakh | 2.51 | 5.92 | 5.92 | 5.92 | 5.92 | 3.79 | 5.92 | 3.91 | — | 2.66 | — | 1.15 | 1.28 |
| Kabiye | 3.48 | 4.87 | 4.87 | 4.87 | 4.87 | 4.74 | 4.87 | 3.28 | — | — | — | 2.98 | 2.71 |
| Kabuverdianu | 1.58 | 1.93 | 1.93 | 1.93 | 1.93 | 1.72 | 1.93 | 1.32 | 1.30 | 1.21 | 0.98 | 1.35 | 1.30 |
| Halh Mongolian | 2.76 | 6.42 | 6.42 | 6.42 | 6.42 | 3.77 | 6.42 | 4.24 | — | 2.72 | — | 1.21 | 1.34 |
| Khmer | 10.26 | 15.33 | 15.33 | 15.33 | 15.33 | 8.88 | 15.33 | 10.22 | — | — | — | 1.62 | 1.87 |
| Kikuyu | 2.52 | 3.44 | 3.44 | 3.44 | 3.44 | 3.29 | 3.44 | 2.36 | — | 1.66 | 1.18 | 2.31 | 2.17 |
| Kinyarwanda | 2.04 | 2.37 | 2.37 | 2.37 | 2.37 | 2.14 | 2.37 | 1.61 | 1.59 | 1.47 | 1.15 | 1.72 | 1.63 |
| Kyrgyz | 2.44 | 5.74 | 5.74 | 5.74 | 5.74 | 3.51 | 5.74 | 3.79 | — | 2.67 | — | 1.16 | 1.66 |
| Kimbundu | 2.02 | 2.33 | 2.33 | 2.33 | 2.33 | 2.13 | 2.33 | 1.64 | 1.58 | 1.43 | 1.12 | 1.64 | 1.54 |
| Northern Kurdish | 2.05 | 2.45 | 2.45 | 2.45 | 2.45 | 2.20 | 2.45 | 1.66 | 1.65 | 1.40 | 0.99 | 1.38 | 1.66 |
| Central Kanuri (Arabic script) | 3.82 | 4.74 | 4.74 | 4.74 | 4.74 | 3.63 | 4.74 | 5.20 | — | — | — | 2.60 | 2.49 |
| Central Kanuri (Latin script) | 2.15 | 2.57 | 2.57 | 2.57 | 2.57 | 2.37 | 2.57 | 1.78 | 1.60 | 1.44 | — | 1.74 | 1.65 |
| Kikongo | 1.93 | 2.17 | 2.17 | 2.17 | 2.17 | 1.99 | 2.17 | 1.61 | 1.44 | 1.37 | 1.12 | 1.58 | 1.48 |
| Korean | 3.18 | 5.07 | 5.07 | 5.07 | 5.07 | 2.38 | 5.07 | 3.86 | — | — | 0.99 | 1.16 | 1.21 |
| Lao | 11.47 | 13.19 | 13.19 | 13.19 | 13.19 | 9.62 | 13.19 | 8.79 | — | — | — | 1.39 | 1.61 |
| Ligurian | 1.84 | 2.29 | 2.29 | 2.29 | 2.29 | 1.98 | 2.29 | 1.57 | 1.50 | 1.43 | 1.09 | 1.65 | 1.59 |
| Limburgish | 1.64 | 2.05 | 2.05 | 2.05 | 2.05 | 1.80 | 2.05 | 1.34 | 1.39 | 1.32 | 1.04 | 1.45 | 1.38 |
| Lingala | 1.79 | 2.03 | 2.03 | 2.03 | 2.03 | 1.86 | 2.03 | 1.47 | 1.37 | 1.26 | 1.08 | 1.52 | 1.26 |
| Lithuanian | 1.89 | 2.45 | 2.45 | 2.45 | 2.45 | 2.21 | 2.45 | 1.63 | 1.53 | 1.42 | 1.04 | 1.17 | 1.25 |
| Lombard | 1.85 | 2.37 | 2.37 | 2.37 | 2.37 | 2.04 | 2.37 | 1.58 | 1.52 | 1.41 | 1.04 | 1.71 | 1.56 |
| Latgalian | 1.99 | 2.39 | 2.39 | 2.39 | 2.39 | 2.20 | 2.39 | 1.67 | 1.62 | 1.48 | 1.02 | 1.57 | 1.51 |
| Luxembourgish | 1.80 | 2.25 | 2.25 | 2.25 | 2.25 | 1.99 | 2.25 | 1.30 | 1.52 | 1.43 | 1.15 | 1.64 | 1.32 |
| Luba-Kasai | 1.89 | 2.13 | 2.13 | 2.13 | 2.13 | 1.94 | 2.13 | 1.50 | 1.44 | 1.31 | 1.09 | 1.54 | 1.43 |
| Ganda | 1.90 | 2.17 | 2.17 | 2.17 | 2.17 | 1.96 | 2.17 | 1.48 | 1.47 | 1.36 | 1.07 | 1.55 | 1.38 |
| Luo | 1.76 | 2.04 | 2.04 | 2.04 | 2.04 | 1.82 | 2.04 | 1.40 | 1.39 | 1.27 | 1.03 | 1.52 | 1.43 |
| Mizo | 1.86 | 2.09 | 2.09 | 2.09 | 2.09 | 1.96 | 2.09 | 1.53 | 1.52 | 1.29 | 1.06 | 1.65 | 1.54 |
| Standard Latvian | 2.10 | 2.54 | 2.54 | 2.54 | 2.54 | 2.35 | 2.54 | 1.76 | 1.68 | 1.56 | 1.05 | 1.23 | 1.29 |
| Magahi | 4.49 | 7.22 | 7.22 | 7.22 | 7.22 | 4.70 | 7.22 | 8.07 | — | — | — | 1.41 | 1.50 |
| Maithili | 4.63 | 7.43 | 7.43 | 7.43 | 7.43 | 4.90 | 7.43 | 8.44 | — | — | — | 1.58 | 1.64 |
| Malayalam | 5.54 | 15.24 | 15.24 | 15.24 | 15.24 | 9.00 | 15.24 | 10.16 | — | — | — | 1.38 | 1.59 |
| Marathi | 4.58 | 7.87 | 7.87 | 7.87 | 7.87 | 5.07 | 7.87 | 8.76 | — | — | — | 1.22 | 1.38 |
| Minangkabau (Arabic script) | 4.32 | 5.25 | 5.25 | 5.25 | 5.25 | 3.97 | 5.25 | 5.71 | — | — | — | 2.02 | 1.99 |
| Minangkabau (Latin script) | 1.77 | 1.97 | 1.97 | 1.97 | 1.97 | 1.77 | 1.97 | 1.40 | 1.39 | 1.25 | 1.09 | 1.31 | 1.25 |
| Macedonian | 1.84 | 5.46 | 5.46 | 5.46 | 5.46 | 2.77 | 5.46 | 3.48 | — | 2.58 | — | 1.17 | 1.24 |
| Maltese | 2.16 | 2.69 | 2.69 | 2.69 | 2.69 | 2.41 | 2.69 | 1.80 | 1.72 | 1.57 | 1.03 | 1.96 | 1.87 |
| Meitei (Bengali script) | 5.84 | 10.22 | 10.22 | 10.22 | 10.22 | 6.71 | 10.22 | 9.06 | — | — | — | 2.56 | 2.59 |
| Mossi | 2.12 | 2.54 | 2.54 | 2.54 | 2.54 | 2.32 | 2.54 | 1.74 | 1.51 | 1.38 | 0.85 | 1.78 | 1.66 |
| Maori | 2.18 | 2.45 | 2.45 | 2.45 | 2.45 | 2.35 | 2.45 | 1.77 | 1.69 | 1.47 | 1.05 | 1.86 | 1.74 |
| Burmese | 8.37 | 16.89 | 16.89 | 16.89 | 16.89 | 11.70 | 16.89 | 11.26 | — | — | — | 1.72 | 2.21 |
| Dutch | 1.46 | 1.97 | 1.97 | 1.97 | 1.97 | 1.59 | 1.97 | 1.28 | 1.40 | 1.32 | 1.13 | 1.14 | 1.18 |
| Norwegian Nynorsk | 1.54 | 1.93 | 1.93 | 1.93 | 1.93 | 1.64 | 1.93 | 1.25 | 1.40 | 1.29 | 1.02 | 1.17 | 1.17 |
| Norwegian Bokmål | 1.50 | 1.86 | 1.86 | 1.86 | 1.86 | 1.86 | 1.86 | 1.23 | 1.37 | 1.27 | 1.01 | 1.07 | 1.10 |
| Nepali | 4.49 | 7.59 | 7.59 | 7.59 | 7.59 | 4.79 | 7.59 | 8.37 | — | — | — | 1.13 | 1.28 |
| Northern Sotho | 2.02 | 2.32 | 2.32 | 2.32 | 2.32 | 2.18 | 2.32 | 1.63 | 1.58 | 1.48 | 1.12 | 1.75 | 1.52 |
| Nuer | 2.83 | 4.23 | 4.23 | 4.23 | 4.23 | 4.00 | 4.23 | 2.79 | — | — | — | 2.62 | 2.44 |
| Nyanja | 2.02 | 2.26 | 2.26 | 2.26 | 2.26 | 2.08 | 2.26 | 1.57 | 1.55 | 1.42 | 1.17 | 1.59 | 1.55 |
| Occitan | 1.66 | 2.07 | 2.07 | 2.07 | 2.07 | 1.83 | 2.07 | 1.47 | 1.40 | 1.38 | 1.14 | 1.50 | 1.31 |
| Odia | 11.59 | 13.38 | 13.38 | 13.38 | 13.38 | 12.48 | 13.38 | 8.94 | — | — | — | 1.45 | 1.56 |

| Language | MBart50 | mT5 | FlanT5 | ByT5 | CANINE | BLOOM | ArabicBERT | MuRIL | UTF-32 | BERT Japanese | SeamlessM4T | NLLB | Qwen |
|---|---|---|---|---|---|---|---|---|---|---|---|---|---|
| Hebrew | 1.12 | 1.22 | — | 1.39 | 0.78 | 2.92 | 1.72 | — | 0.78 | — | 1.24 | 1.24 | 1.48 |
| Hindi | 1.25 | 1.59 | — | 2.55 | 1.00 | 1.28 | — | 1.16 | 1.00 | — | 1.22 | 1.22 | 4.47 |
| Chhattisgarhi | 1.41 | 1.60 | — | 2.46 | 0.97 | 1.44 | — | 1.34 | 0.97 | — | 1.26 | 1.26 | 4.26 |
| Croatian | 1.10 | 1.30 | 2.43 | 1.01 | 0.98 | 1.80 | 1.36 | — | 0.98 | 1.27 | 1.17 | 1.17 | 1.83 |
| Hungarian | 1.18 | 1.26 | 2.99 | 1.16 | 1.05 | 2.07 | 1.40 | 2.31 | 1.05 | — | 1.27 | 1.27 | 2.12 |
| Armenian | 1.38 | 1.58 | — | 2.04 | 1.11 | 4.31 | — | — | 1.11 | — | 1.51 | 1.51 | 5.34 |
| Igbo | 2.12 | 1.79 | 3.17 | 1.21 | 1.02 | 1.72 | 1.50 | — | 1.02 | — | 1.32 | 1.32 | 2.37 |
| Ilocano | 1.61 | 1.61 | 2.82 | 1.21 | 1.21 | 1.90 | 1.55 | 2.01 | 1.21 | 1.55 | 1.33 | 1.33 | 2.03 |
| Indonesian | 0.94 | 1.08 | 2.24 | 1.08 | 1.08 | 0.96 | 1.35 | 1.74 | 1.08 | 1.33 | 0.93 | 0.93 | 1.54 |
| Icelandic | 1.23 | 1.32 | 2.81 | 1.09 | 0.99 | 1.99 | 1.34 | — | 0.99 | — | 1.29 | 1.29 | 2.11 |
| Italian | 1.19 | 1.34 | 2.18 | 1.19 | 1.18 | 1.62 | 1.41 | 1.92 | 1.18 | 1.37 | 1.25 | 1.25 | 1.62 |
| Javanese | 1.15 | 1.21 | 2.21 | 1.04 | 1.04 | 1.40 | 1.36 | 1.74 | 1.04 | 1.29 | 1.03 | 1.03 | 1.72 |
| Japanese | 1.11 | 0.90 | — | 1.27 | 0.44 | 1.81 | 1.01 | — | 0.44 | 0.67 | 1.01 | 1.01 | 1.46 |
| Kabyle | 1.84 | 1.82 | 2.83 | 1.06 | 0.99 | 2.02 | 1.29 | — | 0.99 | — | 1.56 | 1.56 | 2.14 |
| Jingpho | 1.94 | 1.79 | 3.41 | 1.27 | 1.28 | 2.14 | 1.71 | 2.32 | 1.28 | 1.65 | 1.47 | 1.47 | 2.32 |
| Kamba | 1.62 | 1.52 | 2.69 | 1.01 | 0.98 | 1.77 | 1.33 | — | 0.98 | — | 1.28 | 1.28 | 1.99 |
| Kannada | 1.36 | 1.44 | — | 2.83 | 1.05 | 1.31 | — | 1.06 | 1.05 | — | 1.37 | 1.37 | 6.98 |
| Kashmiri (Arabic script) | 1.93 | 2.00 | — | 1.72 | 0.96 | 2.32 | 1.26 | 1.75 | 0.96 | — | 1.81 | 1.81 | 3.48 |
| Kashmiri (Devanagari script) | 1.82 | 1.79 | — | 2.40 | 0.96 | 1.85 | — | 1.75 | 0.96 | — | 1.69 | 1.69 | 4.41 |
| Georgian | 1.34 | 1.55 | — | 2.95 | 1.10 | 4.98 | — | — | 1.10 | — | 1.61 | 1.61 | 5.25 |
| Kazakh | 1.15 | 1.20 | — | 1.89 | 1.03 | 3.23 | — | — | 1.03 | — | 1.18 | 1.18 | 3.02 |
| Kabiye | 2.98 | 2.83 | — | 1.37 | 1.09 | 3.34 | — | — | 1.09 | — | 1.56 | 1.56 | 3.35 |
| Kabuverdianu | 1.35 | 1.28 | 2.21 | 1.02 | 0.99 | 1.51 | 1.25 | 1.81 | 0.99 | 1.29 | 1.28 | 1.28 | 1.70 |
| Halh Mongolian | 1.21 | 1.48 | — | 1.91 | 1.04 | 3.38 | — | — | 1.04 | — | 1.36 | 1.36 | 3.10 |
| Khmer | 1.62 | 1.43 | — | 3.33 | 1.18 | 6.40 | — | — | 1.18 | — | 1.80 | 1.80 | 6.61 |
| Kikuyu | 2.31 | 2.18 | — | 1.30 | 1.17 | 2.48 | 1.56 | — | 1.17 | — | 1.52 | 1.52 | 2.66 |
| Kinyarwanda | 1.72 | 1.51 | 2.76 | 1.13 | 1.11 | 1.58 | 1.54 | 2.15 | 1.11 | 1.50 | 1.30 | 1.30 | 2.12 |
| Kyrgyz | 1.16 | 1.32 | — | 1.88 | 1.02 | 3.02 | — | — | 1.02 | — | 1.25 | 1.25 | 2.74 |
| Kimbundu | 1.64 | 1.48 | 2.91 | 1.11 | 1.11 | 1.81 | 1.55 | 1.99 | 1.11 | 1.52 | 1.35 | 1.35 | 2.10 |
| Northern Kurdish | 1.38 | 1.42 | 2.74 | 1.10 | 1.00 | 2.03 | 1.29 | — | 1.00 | — | 1.44 | 1.44 | 2.16 |
| Central Kanuri (Arabic script) | 2.60 | 2.43 | — | 1.60 | 0.88 | 2.10 | — | 2.37 | 0.88 | — | 2.54 | 2.54 | 3.15 |
| Central Kanuri (Latin script) | 1.74 | 1.58 | 2.82 | 1.11 | 1.05 | 2.00 | — | — | 1.05 | — | 1.55 | 1.55 | 2.16 |
| Kikongo | 1.58 | 1.46 | 3.01 | 1.14 | 1.14 | 1.75 | 1.59 | 1.97 | 1.14 | 1.54 | 1.21 | 1.21 | 1.98 |
| Korean | 1.16 | 1.27 | — | 1.20 | 0.51 | 2.79 | 1.30 | — | 0.51 | — | 1.03 | 1.03 | 1.64 |
| Lao | 1.39 | 1.27 | — | 2.73 | 0.99 | 8.70 | — | — | 0.99 | — | 1.47 | 1.47 | 5.79 |
| Ligurian | 1.65 | 1.69 | 2.54 | 1.17 | 1.10 | 1.81 | 1.38 | 2.05 | 1.10 | — | 1.60 | 1.60 | 1.95 |
| Limburgish | 1.45 | 1.38 | 2.25 | 1.07 | 1.04 | 1.75 | 1.32 | 1.92 | 1.04 | 1.28 | 1.44 | 1.44 | 1.78 |
| Lingala | 1.52 | 1.38 | 2.73 | 1.08 | 1.08 | 1.65 | 1.47 | 1.90 | 1.08 | 1.41 | 1.12 | 1.12 | 1.85 |
| Lithuanian | 1.17 | 1.23 | 2.58 | 1.06 | 1.00 | 1.94 | 1.33 | — | 1.00 | — | 1.18 | 1.18 | 2.06 |
| Lombard | 1.71 | 1.70 | 2.58 | 1.16 | 1.07 | 1.84 | 1.29 | 1.96 | 1.07 | — | 1.61 | 1.61 | 2.00 |
| Latgalian | 1.57 | 1.46 | 2.70 | 1.05 | 0.99 | 1.99 | 1.36 | — | 0.99 | — | 1.42 | 1.42 | 2.14 |
| Luxembourgish | 1.64 | 1.46 | 2.24 | 1.15 | 1.12 | 1.89 | 1.40 | 2.17 | 1.12 | 1.31 | 1.44 | 1.44 | 1.96 |
| Luba-Kasai | 1.54 | 1.37 | 2.48 | 1.08 | 1.08 | 1.68 | 1.44 | 1.89 | 1.08 | 1.41 | 1.21 | 1.21 | 1.92 |
| Ganda | 1.55 | 1.40 | 2.65 | 1.03 | 1.02 | 1.67 | 1.46 | 1.94 | 1.02 | 1.41 | 1.26 | 1.26 | 1.94 |
| Luo | 1.52 | 1.41 | 2.55 | 1.05 | 1.05 | 1.68 | 1.35 | 1.87 | 1.05 | 1.35 | 1.24 | 1.24 | 1.81 |
| Mizo | 1.65 | 1.57 | 2.76 | 1.10 | 1.10 | 1.83 | 1.43 | 1.92 | 1.10 | 1.37 | 1.31 | 1.31 | 1.94 |
| Standard Latvian | 1.23 | 1.30 | 2.78 | 1.11 | 1.02 | 2.08 | 1.35 | — | 1.02 | — | 1.20 | 1.20 | 2.29 |
| Magahi | 1.41 | 1.61 | — | 2.46 | 0.96 | 1.45 | — | 1.34 | 0.96 | — | 1.23 | 1.23 | 4.23 |
| Maithili | 1.58 | 1.74 | — | 2.53 | 0.98 | 1.56 | — | 1.50 | 0.98 | — | 1.24 | 1.24 | 4.42 |
| Malayalam | 1.38 | 1.35 | — | 3.10 | 1.13 | 1.38 | — | 1.18 | 1.13 | — | 1.49 | 1.49 | 7.31 |
| Marathi | 1.22 | 1.52 | — | 2.67 | 1.01 | 1.21 | — | 1.06 | 1.01 | — | 1.26 | 1.26 | 4.65 |
| Minangkabau (Arabic script) | 2.02 | 1.84 | — | 1.74 | 0.96 | 2.58 | 1.13 | — | 0.96 | — | 1.97 | 1.97 | 2.79 |
| Minangkabau (Latin script) | 1.31 | 1.25 | 2.35 | 1.07 | 1.07 | 1.44 | 1.36 | 1.77 | 1.07 | 1.32 | 1.15 | 1.15 | 1.75 |
| Macedonian | 1.17 | 1.29 | — | 1.89 | 1.04 | 2.50 | — | — | 1.04 | — | 1.24 | 1.24 | 2.26 |
| Maltese | 1.96 | 1.69 | 2.94 | 1.16 | 1.11 | 2.25 | 1.44 | — | 1.11 | — | 1.46 | 1.46 | 2.24 |
| Meitei (Bengali script) | 2.56 | 2.21 | — | 2.77 | 1.03 | 2.35 | — | 2.34 | 1.03 | — | 1.73 | 1.73 | 5.64 |
| Mossi | 1.78 | 1.80 | 2.90 | 1.03 | 0.96 | 1.99 | 1.19 | — | 0.96 | — | 1.36 | 1.36 | 2.06 |
| Maori | 1.86 | 1.69 | 3.28 | 1.16 | 1.11 | 2.12 | 1.49 | 2.12 | 1.11 | 1.45 | 1.38 | 1.38 | 2.33 |
| Burmese | 1.72 | 1.56 | — | 3.51 | 1.24 | 10.05 | — | — | 1.24 | — | 1.59 | 1.59 | 8.99 |
| Dutch | 1.14 | 1.17 | 2.19 | 1.11 | 1.11 | 1.71 | 1.38 | 1.91 | 1.11 | 1.33 | 1.19 | 1.19 | 1.58 |
| Norwegian Nynorsk | 1.17 | 1.18 | 2.29 | 1.04 | 1.01 | 1.65 | 1.28 | 1.82 | 1.01 | 1.22 | 1.16 | 1.16 | 1.63 |
| Norwegian Bokmål | 1.07 | 1.12 | 2.24 | 1.03 | 1.01 | 1.62 | 1.26 | 1.79 | 1.01 | 1.18 | 1.10 | 1.10 | 1.55 |
| Nepali | 1.13 | 1.47 | — | 2.56 | 0.96 | 1.17 | — | 1.01 | 0.96 | — | 1.18 | 1.18 | 4.45 |
| Northern Sotho | 1.75 | 1.57 | 2.81 | 1.17 | 1.15 | 1.94 | 1.48 | 2.18 | 1.15 | 1.48 | 1.35 | 1.35 | 2.17 |
| Nuer | 2.62 | 2.42 | — | 1.32 | 1.08 | 2.79 | — | — | 1.08 | — | 1.89 | 1.89 | 3.39 |
| Nyanja | 1.59 | 1.35 | 2.71 | 1.12 | 1.12 | 1.78 | 1.52 | 2.02 | 1.12 | 1.44 | 1.15 | 1.15 | 2.06 |
| Occitan | 1.50 | 1.48 | 2.26 | 1.17 | 1.14 | 1.49 | 1.33 | 1.93 | 1.14 | 1.33 | 1.40 | 1.40 | 1.81 |
| Odia | 1.45 | 3.11 | — | 2.73 | 1.03 | 1.36 | — | 1.21 | 1.03 | — | 1.38 | 1.38 | 9.79 |

| Language | LLAMA | GPT-2 | r50k_base | p50k_base | p50k_edit | cl100k_base | RoBERTa | GottBERT | CamemBERT | PhoBERT | RoCBert | XLM-RoBERTa | M2M100 |
|---|---|---|---|---|---|---|---|---|---|---|---|---|---|
| Pangasinan | 1.50 | 1.66 | 1.66 | 1.66 | 1.66 | 1.57 | 1.66 | 1.27 | 1.25 | 1.11 | 1.00 | 1.29 | 1.23 |
| Eastern Panjabi | 9.44 | 7.90 | 7.90 | 7.90 | 7.90 | 7.87 | 7.90 | 8.47 | — | — | — | 1.57 | 1.68 |
| Papiamento | 1.65 | 1.98 | 1.98 | 1.98 | 1.98 | 1.75 | 1.98 | 1.33 | 1.37 | 1.25 | 1.03 | 1.37 | 1.32 |
| Southern Pashto | 4.27 | 5.39 | 5.39 | 5.39 | 5.39 | 3.83 | 5.39 | 5.37 | — | — | — | 1.38 | 1.40 |
| Western Persian | 3.98 | 5.32 | 5.32 | 5.32 | 5.32 | 3.28 | 5.32 | 5.47 | — | — | — | 1.10 | 1.17 |
| Plateau Malagasy | 2.12 | 2.58 | 2.58 | 2.58 | 2.58 | 2.26 | 2.58 | 1.74 | 1.69 | 1.49 | 1.26 | 1.57 | 1.49 |
| Polish | 1.70 | 2.69 | 2.69 | 2.69 | 2.69 | 1.91 | 2.69 | 1.79 | 1.71 | 1.58 | 1.00 | 1.19 | 1.26 |
| Portuguese | 1.42 | 1.94 | 1.94 | 1.94 | 1.94 | 1.48 | 1.94 | 1.38 | 1.36 | 1.30 | 1.09 | 1.11 | 1.14 |
| Dari | 3.88 | 5.11 | 5.11 | 5.11 | 5.11 | 3.16 | 5.11 | 5.31 | — | — | — | 1.09 | 1.15 |
| Ayacucho Quechua | 1.96 | 2.20 | 2.20 | 2.20 | 2.20 | 2.08 | 2.20 | 1.61 | 1.54 | 1.40 | 1.14 | 1.59 | 1.54 |
| Romanian | 1.70 | 2.48 | 2.48 | 2.48 | 2.48 | 1.88 | 2.48 | 1.69 | 1.54 | 1.46 | 1.13 | 1.24 | 1.29 |
| Rundi | 2.05 | 2.33 | 2.33 | 2.33 | 2.33 | 2.13 | 2.33 | 1.63 | 1.59 | 1.47 | 1.15 | 1.71 | 1.63 |
| Russian | 1.64 | 5.74 | 5.74 | 5.74 | 5.74 | 2.49 | 5.74 | 3.67 | — | 2.71 | 1.03 | 1.17 | 1.22 |
| Sango | 1.95 | 2.23 | 2.23 | 2.23 | 2.23 | 2.08 | 2.23 | 1.54 | 1.50 | 1.32 | 1.02 | 1.66 | 1.53 |
| Sanskrit | 4.59 | 7.94 | 7.94 | 7.94 | 7.94 | 5.00 | 7.94 | 8.60 | — | — | — | 1.43 | 1.69 |
| Santali | 11.92 | 12.86 | 12.86 | 12.86 | 12.86 | 12.80 | 12.86 | 8.56 | — | — | — | — | — |
| Sicilian | 1.81 | 2.27 | 2.27 | 2.27 | 2.27 | 2.01 | 2.27 | 1.60 | 1.43 | 1.37 | 1.06 | 1.58 | 1.53 |
| Shan | 11.85 | 18.76 | 18.76 | 18.76 | 18.76 | 15.05 | 18.76 | 12.51 | — | — | — | 4.43 | 4.63 |
| Sinhala | 7.86 | 12.86 | 12.86 | 12.86 | 12.86 | 8.83 | 12.86 | 8.59 | — | — | — | 1.35 | 1.53 |
| Slovak | 1.82 | 2.52 | 2.52 | 2.52 | 2.52 | 2.14 | 2.52 | 1.65 | 1.60 | 1.46 | 1.02 | 1.18 | 1.24 |
| Slovenian | 1.67 | 2.11 | 2.11 | 2.11 | 2.11 | 1.88 | 2.11 | 1.46 | 1.44 | 1.32 | 1.01 | 1.13 | 1.19 |
| Samoan | 2.14 | 2.57 | 2.57 | 2.57 | 2.57 | 2.29 | 2.57 | 1.69 | 1.63 | 1.50 | 1.09 | 1.92 | 1.80 |
| Shona | 2.01 | 2.29 | 2.29 | 2.29 | 2.29 | 2.13 | 2.29 | 1.58 | 1.58 | 1.44 | 1.18 | 1.63 | 1.58 |
| Sindhi | 4.20 | 5.00 | 5.00 | 5.00 | 5.00 | 4.00 | 5.00 | 5.22 | — | — | — | 1.28 | 1.30 |
| Somali | 2.14 | 2.36 | 2.36 | 2.36 | 2.36 | 2.18 | 2.36 | 1.66 | 1.69 | 1.48 | 1.16 | 1.39 | 1.37 |
| Southern Sotho | 2.07 | 2.34 | 2.34 | 2.34 | 2.34 | 2.21 | 2.34 | 1.64 | 1.63 | 1.48 | 1.18 | 1.78 | 1.60 |
| Spanish | 1.45 | 1.99 | 1.99 | 1.99 | 1.99 | 1.55 | 1.99 | 1.45 | 1.44 | 1.36 | 1.19 | 1.20 | 1.21 |
| Sardinian | 1.82 | 2.26 | 2.26 | 2.26 | 2.26 | 1.99 | 2.26 | 1.53 | 1.48 | 1.40 | 1.16 | 1.61 | 1.51 |
| Serbian | 1.73 | 5.34 | 5.34 | 5.34 | 5.34 | 2.92 | 5.34 | 3.41 | — | 2.45 | — | 1.18 | 1.26 |
| Swati | 2.03 | 2.31 | 2.31 | 2.31 | 2.31 | 2.16 | 2.31 | 1.59 | 1.60 | 1.45 | 1.21 | 1.61 | 1.44 |
| Sundanese | 1.76 | 2.02 | 2.02 | 2.02 | 2.02 | 1.82 | 2.02 | 1.39 | 1.39 | 1.24 | 1.07 | 1.22 | 1.10 |
| Swedish | 1.44 | 1.95 | 1.95 | 1.95 | 1.95 | 1.58 | 1.95 | 1.22 | 1.41 | 1.31 | 1.02 | 1.07 | 1.10 |
| Swahili | 1.86 | 2.13 | 2.13 | 2.13 | 2.13 | 1.95 | 2.13 | 1.49 | 1.42 | 1.32 | 1.06 | 1.16 | 1.20 |
| Silesian | 1.95 | 2.60 | 2.60 | 2.60 | 2.60 | 2.18 | 2.60 | 1.74 | 1.70 | 1.59 | 0.99 | 1.65 | 1.59 |
| Tamil | 5.87 | 15.58 | 15.58 | 15.58 | 15.58 | 7.65 | 15.58 | 10.38 | — | — | — | 1.35 | 1.55 |
| Tamasheq (Latin script) | 1.93 | 2.39 | 2.39 | 2.39 | 2.39 | 2.22 | 2.39 | 1.62 | 1.50 | 1.29 | — | 1.71 | 1.57 |
| Tamasheq (Tifinagh script) | 8.42 | 10.43 | 10.43 | 10.43 | 10.43 | 10.13 | 10.43 | 6.95 | — | — | — | — | — |
| Tatar | 2.53 | 5.82 | 5.82 | 5.82 | 5.82 | 3.75 | 5.82 | 3.84 | — | — | — | 1.81 | 1.54 |
| Telugu | 10.71 | 13.09 | 13.09 | 13.09 | 13.09 | 8.34 | 13.09 | 8.73 | — | — | — | 1.33 | — |
| Tajik | 2.70 | 6.09 | 6.09 | 6.09 | 6.09 | 3.64 | 6.09 | 4.00 | — | 2.82 | — | 2.14 | 2.06 |
| Tagalog | 2.00 | 2.28 | 2.28 | 2.28 | 2.28 | 2.06 | 2.28 | 1.60 | 1.67 | 1.45 | 1.27 | 1.43 | 1.43 |
| Thai | 4.35 | 9.05 | 9.05 | 9.05 | 9.05 | 4.39 | 9.05 | 6.59 | — | 2.83 | — | 1.08 | 1.27 |
| Tigrinya | 7.47 | 7.88 | 7.88 | 7.88 | 7.88 | 7.80 | 7.88 | 5.25 | — | — | — | 1.97 | 1.91 |
| Tok Pisin | 1.95 | 2.21 | 2.21 | 2.21 | 2.21 | 2.04 | 2.21 | 1.55 | 1.66 | 1.45 | 1.25 | 1.73 | 1.65 |
| Tswana | 2.12 | 2.39 | 2.39 | 2.39 | 2.39 | 2.28 | 2.39 | 1.68 | 1.67 | 1.55 | 1.21 | 1.85 | 1.68 |
| Tsonga | 2.16 | 2.45 | 2.45 | 2.45 | 2.45 | 2.26 | 2.45 | 1.70 | 1.70 | 1.46 | 1.19 | 1.79 | 1.69 |
| Turkmen | 2.23 | 2.82 | 2.82 | 2.82 | 2.82 | 2.40 | 2.82 | 1.76 | 1.78 | 1.62 | 1.11 | 1.78 | 1.71 |
| Tumbuka | 2.46 | 2.78 | 2.78 | 2.78 | 2.78 | 2.57 | 2.78 | 1.93 | 1.85 | 1.67 | 1.34 | 1.92 | 1.88 |
| Turkish | 2.09 | 2.43 | 2.43 | 2.43 | 2.43 | 1.91 | 2.43 | 1.61 | 1.65 | 1.51 | — | 1.04 | 1.15 |
| Twi | 2.01 | 2.62 | 2.62 | 2.62 | 2.62 | 2.51 | 2.62 | 1.80 | 1.57 | 1.38 | — | 1.88 | 1.74 |
| Central Atlas Tamazight | 8.86 | 10.39 | 10.39 | 10.39 | 10.39 | 10.04 | 10.39 | 6.92 | — | — | — | — | — |
| Uyghur | 4.89 | 7.16 | 7.16 | 7.16 | 7.16 | 5.19 | 7.16 | 6.44 | — | — | — | 1.41 | 3.00 |
| Ukrainian | 1.72 | 5.75 | 5.75 | 5.75 | 5.75 | 3.00 | 5.75 | 3.69 | — | 2.58 | — | 1.21 | 1.28 |
| Umbundu | 1.89 | 2.24 | 2.24 | 2.24 | 2.24 | 2.01 | 2.24 | 1.53 | 1.48 | 1.36 | 1.05 | 1.57 | 1.49 |
| Urdu | 4.37 | 6.30 | 6.30 | 6.30 | 6.30 | 4.39 | 6.30 | 5.74 | — | — | — | 1.23 | 1.30 |
| Northern Uzbek | 2.03 | 2.30 | 2.30 | 2.30 | 2.30 | 2.17 | 2.30 | 1.63 | 1.59 | 1.48 | 1.19 | 1.33 | 1.37 |
| Venetian | 1.56 | 2.00 | 2.00 | 2.00 | 2.00 | 1.70 | 2.00 | 1.38 | 1.34 | 1.23 | — | 1.36 | 1.31 |
| Vietnamese | 2.92 | 4.54 | 4.54 | 4.54 | 4.54 | 2.45 | 4.54 | 3.06 | — | 0.83 | 0.98 | 1.18 | 1.15 |
| Waray | 2.02 | 2.38 | 2.38 | 2.38 | 2.38 | 1.95 | 2.38 | 1.61 | 1.66 | 1.42 | 1.25 | 1.55 | 1.45 |
| Wolof | 1.80 | 2.14 | 2.14 | 2.14 | 2.14 | 1.92 | 2.14 | 1.49 | 1.43 | 1.28 | 0.93 | 1.60 | 1.40 |
| Xhosa | 1.97 | 2.26 | 2.26 | 2.26 | 2.26 | 2.06 | 2.26 | 1.57 | 1.57 | 1.40 | 1.13 | 1.50 | 1.37 |
| Eastern Yiddish | 4.57 | 6.63 | 6.63 | 6.63 | 6.63 | 5.57 | 6.63 | 6.34 | — | — | — | 1.58 | 1.61 |
| Yoruba | 2.70 | 3.89 | 3.89 | 3.89 | 3.89 | 2.96 | 3.89 | 2.63 | — | 1.66 | 0.88 | 2.27 | 1.74 |
| Yue Chinese | 2.11 | 3.09 | 3.09 | 3.09 | 3.09 | 2.12 | 3.09 | 2.78 | — | — | 0.36 | 0.93 | 1.03 |
| Chinese (Simplified) | 2.00 | 3.21 | 3.21 | 3.21 | 3.21 | 1.91 | 3.21 | 2.93 | — | — | 0.39 | 0.97 | 1.05 |
| Chinese (Traditional) | 2.16 | 3.16 | 3.16 | 3.16 | 3.16 | 2.18 | 3.16 | 2.83 | — | — | 0.36 | 0.96 | 1.06 |
| Standard Malay | 1.83 | 2.05 | 2.05 | 2.05 | 2.05 | 1.62 | 2.05 | 1.42 | 1.45 | 1.28 | 1.15 | 0.95 | 1.00 |
| Zulu | 2.09 | 2.41 | 2.41 | 2.41 | 2.41 | 2.20 | 2.41 | 1.65 | 1.64 | 1.47 | 1.20 | 1.55 | 1.35 |

| Language | MBart50 | mT5 | FlanT5 | ByT5 | CANINE | BLOOM | ArabicBERT | MuRIL | UTF-32 | BERT Japanese | SeamlessM4T | NLLB | Qwen |
|---|---|---|---|---|---|---|---|---|---|---|---|---|---|
| Pangasinan | 1.29 | 1.22 | 2.18 | 1.00 | 1.00 | 1.45 | 1.24 | 1.54 | 1.00 | 1.21 | 1.11 | 1.11 | 1.56 |
| Eastern Panjabi | 1.57 | 2.11 | — | 2.59 | 1.01 | 1.43 | — | 1.35 | 1.01 | — | 1.50 | 1.50 | 7.30 |
| Papiamento | 1.37 | 1.36 | 2.28 | 1.08 | 1.05 | 1.54 | 1.25 | 1.80 | 1.05 | 1.30 | 1.27 | 1.27 | 1.73 |
| Southern Pashto | 1.38 | 1.64 | — | 1.66 | 0.95 | 2.55 | — | — | 0.95 | — | 1.45 | 1.45 | 2.87 |
| Western Persian | 1.10 | 1.34 | — | 1.70 | 0.94 | 1.78 | 1.11 | 1.62 | 0.94 | — | 1.13 | 1.13 | 2.60 |
| Plateau Malagasy | 1.57 | 1.59 | 3.00 | 1.26 | 1.22 | 2.07 | 1.64 | 2.33 | 1.22 | 1.59 | 1.39 | 1.39 | 2.23 |
| Polish | 1.19 | 1.31 | 2.82 | 1.13 | 1.06 | 2.14 | 1.52 | — | 1.06 | — | 1.37 | 1.37 | 1.76 |
| Portuguese | 1.11 | 1.29 | 2.21 | 1.12 | 1.09 | 1.12 | 1.30 | 1.88 | 1.09 | 1.24 | 1.17 | 1.17 | 1.45 |
| Dari | 1.09 | 1.31 | — | 1.63 | 0.92 | 1.64 | 1.09 | 1.58 | 0.92 | — | 1.11 | 1.11 | 2.50 |
| Ayacucho Quechua | 1.59 | 1.42 | 2.59 | 1.08 | 1.07 | 1.83 | 1.47 | 1.95 | 1.07 | 1.42 | 1.28 | 1.28 | 2.06 |
| Romanian | 1.24 | 1.37 | 1.50 | 1.19 | 1.13 | 1.91 | 1.33 | — | 1.13 | — | 1.35 | 1.35 | 1.86 |
| Rundi | 1.71 | 1.52 | 2.78 | 1.12 | 1.12 | 1.64 | 1.54 | 2.13 | 1.12 | 1.50 | 1.33 | 1.33 | 2.11 |
| Russian | 1.17 | 1.27 | — | 1.98 | 1.09 | 2.48 | 2.50 | — | 1.09 | — | 1.34 | 1.34 | 1.75 |
| Sango | 1.66 | 1.63 | 3.14 | 1.12 | 1.09 | 1.80 | 1.45 | 2.05 | 1.09 | 1.49 | 1.39 | 1.39 | 2.04 |
| Sanskrit | 1.43 | 1.65 | — | 2.63 | 0.98 | 1.63 | — | 1.21 | 0.98 | — | 1.40 | 1.40 | 4.58 |
| Santali | — | — | — | 2.79 | 1.06 | 12.71 | — | — | 1.06 | — | 2.49 | 2.49 | 8.99 |
| Sicilian | 1.58 | 1.53 | 2.46 | 1.11 | 1.05 | 1.80 | 1.41 | 1.84 | 1.05 | — | 1.44 | 1.44 | 1.95 |
| Shan | 4.43 | 3.28 | — | 3.94 | 1.42 | 12.06 | — | — | 1.42 | — | 1.94 | 1.94 | 10.51 |
| Sinhala | 1.35 | 1.66 | — | 2.64 | 1.00 | 8.21 | — | — | 1.00 | — | 1.68 | 1.68 | 7.02 |
| Slovak | 1.18 | 1.30 | 2.74 | 1.09 | 1.00 | 2.01 | 1.35 | — | 1.00 | — | 1.21 | 1.21 | 2.08 |
| Slovenian | 1.13 | 1.20 | 2.42 | 1.02 | 1.00 | 1.81 | 1.37 | — | 1.00 | 1.30 | 1.17 | 1.17 | 1.87 |
| Samoan | 1.92 | 1.92 | 3.09 | 1.22 | 1.16 | 2.13 | 1.57 | 2.22 | 1.16 | 1.55 | 1.60 | 1.60 | 2.26 |
| Shona | 1.63 | 1.35 | 2.79 | 1.12 | 1.12 | 1.80 | 1.55 | 2.06 | 1.12 | 1.48 | 1.23 | 1.23 | 2.11 |
| Sindhi | 1.28 | 1.74 | — | 1.60 | 0.91 | 2.51 | — | 1.22 | 0.91 | — | 1.33 | 1.33 | 2.87 |
| Somali | 1.39 | 1.48 | 3.06 | 1.14 | 1.14 | 2.03 | 1.52 | 2.05 | 1.14 | 1.52 | 1.39 | 1.39 | 2.16 |
| Southern Sotho | 1.78 | 1.59 | 2.92 | 1.21 | 1.20 | 1.96 | 1.61 | 2.16 | 1.20 | 1.54 | 1.39 | 1.39 | 2.19 |
| Spanish | 1.20 | 1.31 | 2.23 | 1.21 | 1.19 | 1.21 | 1.38 | 1.98 | 1.19 | 1.41 | 1.24 | 1.24 | 1.52 |
| Sardinian | 1.61 | 1.57 | 2.46 | 1.19 | 1.16 | 1.73 | 1.38 | 1.98 | 1.16 | 1.36 | 1.44 | 1.44 | 1.97 |
| Serbian | 1.18 | 1.30 | — | 1.80 | 0.99 | 2.57 | — | — | 0.99 | — | 1.24 | 1.24 | 2.34 |
| Swati | 1.61 | 1.41 | 2.80 | 1.12 | 1.13 | 1.83 | 1.55 | 2.09 | 1.13 | 1.52 | 1.28 | 1.28 | 2.14 |
| Sundanese | 1.22 | 1.22 | 2.32 | 1.05 | 1.04 | 1.48 | 1.33 | 1.80 | 1.04 | 1.31 | 1.04 | 1.04 | 1.80 |
| Swedish | 1.07 | 1.11 | 2.22 | 1.04 | 1.01 | 1.65 | 1.21 | 1.90 | 1.01 | 1.20 | 1.13 | 1.13 | 1.57 |
| Swahili | 1.16 | 1.25 | 2.66 | 1.05 | 1.05 | 1.24 | 1.45 | 1.86 | 1.05 | 1.43 | 1.13 | 1.13 | 1.93 |
| Silesian | 1.65 | 1.57 | 2.87 | 1.10 | 1.04 | 2.16 | 1.52 | — | 1.04 | — | 1.52 | 1.52 | 2.09 |
| Tamil | 1.35 | 1.26 | — | 3.17 | 1.17 | 1.27 | — | 1.06 | 1.17 | — | 1.42 | 1.42 | 6.15 |
| Tamasheq (Latin script) | 1.71 | 1.64 | 2.55 | 1.01 | 0.95 | 1.90 | — | — | 0.95 | — | 1.52 | 1.52 | 1.99 |
| Tamasheq (Tifinagh script) | — | 3.59 | — | 2.29 | 0.94 | 7.74 | — | — | 0.94 | — | 2.43 | 2.43 | 5.37 |
| Tatar | 1.81 | 1.41 | — | 1.89 | 1.01 | 3.15 | — | — | 1.01 | — | 1.21 | 1.21 | 2.88 |
| Telugu | 1.33 | 1.42 | — | 2.68 | 1.01 | 1.33 | — | 1.21 | 1.01 | — | 1.34 | 1.34 | 7.06 |
| Tajik | 2.14 | 1.62 | — | 2.01 | 1.11 | 3.29 | 2.39 | — | 1.11 | — | 1.57 | 1.57 | 2.90 |
| Tagalog | 1.43 | 1.46 | 2.85 | 1.26 | 1.26 | 1.85 | 1.56 | 2.08 | 1.26 | 1.60 | 1.34 | 1.34 | 2.04 |
| Thai | 1.08 | 0.99 | — | 2.75 | 0.96 | 4.63 | — | — | 0.96 | — | 1.52 | 1.52 | 2.59 |
| Tigrinya | 1.97 | 2.03 | — | 1.75 | 0.69 | 5.16 | — | — | 0.69 | — | 1.44 | 1.44 | 4.24 |
| Tok Pisin | 1.73 | 1.65 | 2.76 | 1.28 | 1.28 | 1.92 | 1.61 | 2.10 | 1.28 | 1.57 | 1.39 | 1.39 | 2.02 |
| Tswana | 1.85 | 1.68 | 3.01 | 1.25 | 1.25 | 2.02 | 1.62 | 2.25 | 1.25 | 1.57 | 1.45 | 1.45 | 2.26 |
| Tsonga | 1.79 | 1.61 | 3.13 | 1.20 | 1.20 | 2.01 | 1.65 | 2.19 | 1.20 | 1.64 | 1.30 | 1.30 | 2.23 |
| Turkmen | 1.78 | 1.68 | 2.87 | 1.17 | 1.06 | 2.19 | 1.44 | — | 1.06 | — | 1.36 | 1.36 | 2.20 |
| Tumbuka | 1.92 | 1.61 | 3.29 | 1.32 | 1.30 | 2.19 | 1.79 | — | 1.30 | — | 1.43 | 1.43 | 2.51 |
| Turkish | 1.04 | 1.12 | 2.67 | 1.12 | 1.03 | 1.96 | 1.45 | — | 1.03 | — | 1.14 | 1.14 | 1.61 |
| Twi | 1.88 | 1.71 | 2.85 | 1.05 | 0.98 | 1.81 | — | — | 0.98 | 1.40 | 1.25 | 1.25 | 2.15 |
| Central Atlas Tamazight | — | 3.48 | — | 2.28 | 0.89 | 7.69 | — | — | 0.89 | — | 2.06 | 2.06 | 5.09 |
| Uyghur | 1.41 | 2.57 | — | 1.97 | 1.07 | 3.67 | — | — | 1.07 | — | 1.40 | 1.40 | 3.74 |
| Ukrainian | 1.21 | 1.33 | — | 1.86 | 1.02 | 2.75 | 2.35 | — | 1.02 | — | 1.28 | 1.28 | 2.51 |
| Umbundu | 1.57 | 1.47 | 2.72 | 1.05 | 1.01 | 1.74 | 1.46 | 1.94 | 1.01 | 1.33 | 1.29 | 1.29 | 1.97 |
| Urdu | 1.23 | 1.52 | — | 1.76 | 0.99 | 1.36 | 1.45 | 1.26 | 0.99 | — | 1.30 | 1.30 | 3.19 |
| Northern Uzbek | 1.33 | 1.38 | 2.80 | 1.13 | 1.13 | 1.98 | 1.58 | 2.12 | 1.13 | 1.53 | 1.32 | 1.32 | 2.15 |
| Venetian | 1.36 | 1.36 | 2.21 | 1.06 | 1.01 | 1.57 | 1.24 | 1.84 | 1.01 | 1.23 | 1.29 | 1.29 | 1.68 |
| Vietnamese | 1.18 | 1.95 | — | 1.39 | 1.05 | 1.27 | 1.38 | — | 1.05 | — | 1.18 | 1.18 | 1.41 |
| Waray | 1.55 | 1.45 | 2.66 | 1.25 | 1.25 | 1.80 | 1.60 | 2.15 | 1.25 | 1.52 | 1.36 | 1.36 | 1.93 |
| Wolof | 1.60 | 1.44 | 2.62 | 1.00 | 0.96 | 1.68 | 1.28 | 1.93 | 0.96 | 1.26 | 1.31 | 1.31 | 1.89 |
| Xhosa | 1.50 | 1.35 | 2.73 | 1.06 | 1.06 | 1.67 | 1.52 | 2.05 | 1.06 | 1.45 | 1.21 | 1.21 | 2.04 |
| Eastern Yiddish | 1.58 | 1.66 | — | 1.94 | 1.08 | 4.42 | 2.41 | — | 1.08 | — | 1.69 | 1.69 | 2.77 |
| Yoruba | 2.27 | 2.06 | — | 1.28 | 0.97 | 1.64 | 1.24 | — | 0.97 | — | 1.52 | 1.52 | 2.69 |
| Yue Chinese | 0.93 | 0.95 | — | 0.87 | 0.31 | 0.93 | — | — | 0.31 | 0.55 | 1.05 | 1.05 | 1.17 |
| Chinese (Simplified) | 0.97 | 0.92 | — | 0.93 | 0.34 | 0.95 | — | — | 0.34 | 0.55 | 1.11 | 1.11 | 1.07 |
| Chinese (Traditional) | 0.96 | 0.98 | — | 0.89 | 0.32 | 0.97 | — | — | 0.32 | 0.57 | 1.08 | 1.08 | 1.21 |
| Standard Malay | 0.95 | 1.11 | 2.32 | 1.12 | 1.11 | 1.07 | 1.39 | 1.80 | 1.11 | 1.36 | 0.96 | 0.96 | 1.61 |
| Zulu | 1.55 | 1.40 | 2.84 | 1.12 | 1.12 | 1.76 | 1.62 | 2.15 | 1.12 | 1.54 | 1.24 | 1.24 | 2.18 |

