# OpenReview forum: "Language Model Tokenizers Introduce Unfairness Between Languages"
_NeurIPS.cc/2023/Conference — NeurIPS 2023 poster_

### Official Review · Reviewer_okDv · 2023-07-04

**Soundness:** 3 good
**Presentation:** 3 good
**Contribution:** 3 good
**Rating:** 6
**Confidence:** 4

**Summary:**

The paper studies the discrepancy in the tokenization length in different languages. It shows the unfairness of utilizing tokenization in different languages due to the cost, latency, and long-term language dependencies. The paper evaluates the unfairness between different tokenization strategies and model architectures. The paper's motivation is clear, and it provides an extensive analysis of multiple languages, showing the significance of the study and why it is essential in research and practical perspectives.

**Strengths:**

**Originality**
- The paper introduces an important issue on utilizing LLM in different languages and investigates the aspects of cost, latency, and long-dependency. The problem formulation is important for multilingual researchers and practitioners in deciding what tokenization strategies are cost-effective.

**Quality / Significance**
- The paper shows a comprehensive analysis of how the different tokenization on different languages was done in different tokenization strategies (byte-based, Unicode-based, and subword-based) on the wide range of pre-trained LLM (on encoder-only, encoder-decoder, and decoder-only) and provides the quantitative evidence on the processing time of each language compared to English.

**Clarity**
- The paper is well-written, and the examples (in Japanese and Shan) give a high-level idea of why the problem introduced in the paper is important.

**Weaknesses:**

- Besides the aspects mentioned by the paper, it is unclear if the choice of the tokenization strategies would impact the performance of LLM in particular languages.
- The suggestion of "building a multilingually fair parallel corpus" is vague. It would be great if the authors could give a more straightforward and practical solution.


**Questions:**

**Questions**

- Does the tokenization lengths impact the performance on languages (e.g., Shan)?

**Typo**
Line 177: XML-R => XLM-R

**Limitations:**

Yes.

---

> ### Author Rebuttal · Authors · 2023-08-05
>
> > Besides the aspects mentioned by the paper, it is unclear if the choice of the tokenization strategies would impact the performance of LLM in particular languages.
>
> Tokenization does also affect downstream performance. In the Related Works section, we mention the work by Zhang et al. (2022), which shows that a balanced tokenizer corpus results in better translation performance. The works of Hofmann et al. (2021, 2022) also show that larger tokens, as well as tokenization informed by morphological structures, results in better downstream performance. We will add them to our Related Works section and will discuss further the downstream implications of the tokenization disparities.
>
> > The suggestion of "building a multilingually fair parallel corpus" is vague. It would be great if the authors could give a more straightforward and practical solution.
>
> In order to build a multilingually fair tokenizer one first needs a parallel corpus to use in order to ensure that the conent against which tokenization lengths are measured is indeed comparable. FLORES-200 is big enough for evaluation of tokenizers, but is too small for training ones. However, building a large parallel corpus for so many languages is extremely difficult. In our "On the development of a multilingually fair tokenizer" comment we offer a way to approximate such a multulingual parallel corpus via many bilingual parallel corpora.
>
> > Does the tokenization lengths impact the performance on languages (e.g., Shan)?
>
> Decoupling the effects of the tokenizations and the amount of data for a specific language is not possible. However, a recent work by Liu at el. (2023) showed that the performance of language models decreases as the input sizes grow. Hence, it is very likely that the longer tokenizations for some languages (e.g. Shan) do hurt the performance of that language, keeping everything else the same.
>
> --
>
> Liu, Nelson F., et al. "Lost in the middle: How language models use long contexts." arXiv preprint arXiv:2307.03172. (2023)
>
> Zhang, Shiyue, et al. “How robust is neural machine translation to language imbalance in multilingual tokenizer training?” Biennial Conference of the Association for Machine Translation in the Americas (2022)
>
> Hofmann, Valentin, et al. “Superbizarre Is Not Superb: Derivational Morphology Improves BERT’s Interpretation of Complex Words”. Annual Meeting of the Association for Computational Linguistics (2021)
>
> Hofmann, Valentin, et al. “An Embarrassingly Simple Method to Mitigate Undesirable Properties of Pretrained Language Model Tokenizers”. Annual Meeting of the Association for Computational Linguistics (2022)

---

> > ### Comment · Reviewer_okDv · 2023-08-22
> >
> > Thank you for the comments. I will keep my score unchanged.

---

### Official Review · Reviewer_qshy · 2023-07-06

**Soundness:** 4 excellent
**Presentation:** 4 excellent
**Contribution:** 3 good
**Rating:** 7
**Confidence:** 4

**Summary:**

Tokenization is a crucial, yet underappreciated component of language models. This paper presents a welcome investigation into the effects of tokenization choices across languages. The paper introduces the notion of premium for language A relative to B which measures the ratio of the average number of tokens for translations of the same sentence in the two languages. The authors show that the premium relative to English can be over 10 for certain low resource languages.

The paper then highlights the negative effects of having a high premium, which include: cost, latency, and the ability to fit less content in the fixed context window of an LM. Based on these findings, the authors make a case for developing multilingually fair tokenizers where the premiums are close to one across languages.

**Strengths:**

1. Tokenization is, in my view, an underappreciated problem and this work provides an insightful analysis on the effects of the tokenizer selection.
2. The discrepancies presented in the paper across languages are surprisingly large. As such, this paper may stimulate new research in the field to address the discrepancies.
3. The paper is very well written and a pleasure to read.


**Weaknesses:**

1. The paper doesn’t discuss how to achieve the goal of training multilingually fair tokenizers, apart from providing an overview of previous approaches to developing multilingual tokenizers. An obvious solution is to reserve more tokens for the languages with high premiums. However, this would mean that the number of tokens for other languages would need to be reduced to keep the vocabulary size fixed. This in turn will likely hurt the performance on high-resource languages such as English and it may even end up hurting the performance for the rest of the languages if there is less cross-lingual transfer from high-resource languages (NB: The paper states that “multilingual models struggle to deliver on the promises of deep transfer learning” but I’m not aware of works suggesting that there’s zero transfer happening across languages.)
2. The paper lacks discussion on the potential negative effects of using a fair tokenizer (such as the one presented above) and discussion on the potential means of developing fair tokenizers. Yet, it argues strongly why the switch to fair tokenizers is necessary. This makes the paper seem a bit like a position paper (which doesn’t reduce the value of the paper in my view but makes me wonder a bit whether NeurIPS is an ideal venue for it).
3. An obvious explanation for the tokenization length differences is the widely varying amounts of available training datasets across languages. However, another possible explanation is inherent differences between languages in terms of characteristics such as morphological richness of a language. Discussion of the different explanations seems to be missing from the paper.


**Questions:**

Could you elaborate on the following sentence “training a model from scratch with this fair tokenizer is necessary, despite potential suboptimal performance in individual languages due to vocabulary limitations”? What are the potential suboptimalities you’re referring to? Are you arguing for using a fair tokenizer regardless of any potential negative side effects?

**Limitations:**

Yes.

---

> ### Author Rebuttal · Authors · 2023-08-05
>
> >The paper doesn’t discuss how to achieve the goal of training multilingually fair tokenizers, apart from providing an overview of previous approaches to developing multilingual tokenizers. An obvious solution is to reserve more tokens for the languages with high premiums.
>
> Please see our "On the development of a multilingually fair tokenizer" comment, where we discuss the development of a multilingually fair tokenizer. We also discuss why giving quotas to different languages will not work: languages sharing the same script also partially share tokens.
>
> > This in turn will likely hurt the performance on high-resource languages such as English and it may even end up hurting the performance for the rest of the languages if there is less cross-lingual transfer from high-resource languages.
>
> The additional cost for English would be much smaller than the total benefit for the rest of the languages. That is because increasing the vocabulary size has diminishing returns: the additional tokens correspond to increasingly rare (parts of) words. Therefore, removing rarely used English (sub)words and replacing them with frequently used (sub)words in other languages would likely be of a net befit overall. Furthermore, cross-lingual transfer can be bi-directional: there can be information present in the data in the less-resourced language that is not present in the high-resourced language (for example, cultural, legal or historical knowledge). Therefore, more fair tokenization can also improve cross-lingual transfer in the opposite direction.
>
> > The paper lacks discussion on the potential negative effects of using a fair tokenizer (such as the one presented above) and discussion on the potential means of developing fair tokenizers. Yet, it argues strongly why the switch to fair tokenizers is necessary. This makes the paper seem a bit like a position paper (which doesn’t reduce the value of the paper in my view but makes me wonder a bit whether NeurIPS is an ideal venue for it).
>
> We are not aware of other negative effects of using a fair tokenizer, apart from the above-mentioned issue of slightly increasing the tokenization lengths of the most tokenisation-efficient language. Moreover, in the fairness literature, reducing the preferential treatment of a dominant group to balance the treatment of all groups is typically considered a positive effect rather than a limitation.
>
> We do not consider our work as a position paper, especially because it is based on a comprehensive empirical evaluation. We do say that if one doesn't want to charge users of different languages more, reduce their effective context size and increase their processing time, then they would need to switch to a more fair tokenizer. However, this is backed by our results and analysis of the underlying technical factors, rather than simply our views or opinions, as customary for position papers.
>
> > An obvious explanation for the tokenization length differences is the widely varying amounts of available training datasets across languages. However, another possible explanation is inherent differences between languages in terms of characteristics such as morphological richness of a language. Discussion of the different explanations seems to be missing from the paper.
>
> The widely varying amounts of available training datasets across languages are certainly the main culprit. However, as mentioned above, even if we use a perfectly balanced dataset, parity will not be achieved because some languages share tokens while others do not. Assessing morphological differences between languages is a challenging task as more morphological richness does not necessarily imply that more tokens would be necessary. As the information content of all languages is the same (Coupé et al., 2019), we argue that similar tokenization lengths should be achievable. We looked into the absolute number of tokens needed to encode the FLORES-200 dataset across several languages and tokenizers targeting them. The numbers cannot be directly compared, though, as the tokenizers might have different vocabulary sizes and might be trained differently (e.g. BPE vs SentencePiece). Still, the differences are much smaller than the fairness premiums we observe. That indicates that morphological differences cannot explain the drastic variations reported in the paper. Thank you for highlighting this question; we will incorporate it in the manuscript!
>
> |Language|Tokenizer|Number of tokens for FLORES-200|
> |----|----|----|
> |Standard Arabic|ArabicBERT|52834|
> |German|GottBERT|58508|
> |English|RoBERTa|52567|
> |French|CamemBERT |67031|
> |Hindi|MuRIL|62712|
> |Japanese|BERT Japanese|69209|
> |Chinese (Simplified) | RoCBert |83317|
> |Vietnamese|PhoBERT|69628|
>
> > Could you elaborate on the following sentence “training a model from scratch with this fair tokenizer is necessary, despite potential suboptimal performance in individual languages due to vocabulary limitations”? What are the potential suboptimalities you’re referring to?
>
> We mean that one cannot get away from having to retrain the model using the fair tokenizer. In other words, it is not possible to post factum make the tokenizer of a pre-trained model fair. The suboptimal performance can stem from representing a language in the tokenization, but then training the model with little data from this language. In such a case the tokenizer would be fair but the model would not perform well in the underrepresented language. However, this problem is resolvable by one ensuring sufficient training data for all languages with respect to which the model should be fair. We agree that the sentence could be better worded and will edit it.
>
> --
>
> Christophe Coupé et al. “Different languages, similar encoding efficiency: Comparable information rates across the human communicative niche”. Science Advances (2019)

---

> > ### Comment · Reviewer_qshy · 2023-08-19
> >
> > Thank you for your responses.
> >
> > > even if we use a perfectly balanced dataset, parity will not be achieved because some languages share tokens while others do not. Assessing morphological differences between languages is a challenging task as more morphological richness does not necessarily imply that more tokens would be necessary. As the information content of all languages is the same (Coupé et al., 2019), we argue that similar tokenization lengths should be achievable.
> >
> > I'm not convinced that token sharing is the main cause for not achieving parity even when using balanced dataset. Two other causes that could be even more significant and would be useful to discuss in the paper:
> > 1. **Morphological differences:** To give an example, consider a tokenizer trained on a balanced set of English and German nouns and their plural forms. Let's fix the vocabulary size such that every English plural noun is split into at most 2 tokens. Now, we would very likely see a premium for German since a significant fraction of German nouns require umlaut when pluralized (e.g. `Stuhl` (chair) -> `Stühle` (chairs)) which makes token sharing across the singular and plural forms within the same language more difficult.
> > 2. **Number of characters:** For instance, Korean encoded as Hangul Syllables has ~11k Unicode characters whereas the Latin (ASCII) has only ~100.
> >
> > > increasing the vocabulary size has diminishing returns: the additional tokens correspond to increasingly rare (parts of) words. Therefore, removing rarely used English (sub)words and replacing them with frequently used (sub)words in other languages would likely be of a net befit overall.
> >
> > I agree that this is likely the case due to the diminishing returns. The paper would be much stronger if it experimentally supported this point by actually training a fairer tokenizer (as also proposed by several other reviewers) and measuring the net benefit.
> >
> > Having said that, even in it's current form, I find this a strong paper which is likely to inspire several future works in the NLP community.

---

> > > ### Author Response · Authors · 2023-08-21
> > >
> > > Thank you so much for your feedback and for your support of our work!
> > >
> > > You are right, morphological differences do complicate the task significantly, with `Stuhl` being a great example.
> > >
> > > As for Korean, none of the authors is a speaker of Korean, but to the best of our understanding, there are two ways to encode Korean in Unicode. One is Hangul Syllables which do indeed have reserved ~11k Unicode codepoints. However, as Korean syllables are compositional, one can also encode it via [Hangul Jamo](https://en.wikipedia.org/wiki/Hangul_Jamo_(Unicode_block)). There are 256 Jamo codepoints, but most of them seem to not be used in modern Korean. According to the article, only 67 Jamo are used in modern Korean, which means that 67 Unicode codepoints should be sufficient to represent most modern content. Choosing how to normalize the Unicode string (Syllables or Jamo) would hence have a great impact on the resulting tokenization, which further illustrates that one has to be very careful when approaching multilingual tokenization.
> > >
> > > > I agree that this is likely the case due to the diminishing returns. The paper would be much stronger if it experimentally supported this point by actually training a fairer tokenizer (as also proposed by several other reviewers) and measuring the net benefit.
> > >
> > > Perhaps the reviewer would be interested in seeing our response to Reviewer Qimt, where we show that with only one-third of the vocabulary of the ChatGPT/GPT-4 tokenizer, English sequences will become just 10% longer.

---

### Official Review · Reviewer_Qimt · 2023-07-07

**Soundness:** 3 good
**Presentation:** 3 good
**Contribution:** 2 fair
**Rating:** 4
**Confidence:** 4

**Summary:**

This paper investigates the disparities between languages caused by tokenization policies used in large language models (LLMs). The authors compare the numbers of tokens needed to represent the translations of the same sentence in different languages and observe that the number of tokens needed in one language can be an order of magnitude larger than that in the target language (mostly English). They argue that these disparities have significant real-world implications such as the increased cost and performance degradations in using LLMs.

**Strengths:**

- I think investigating the disparities of the cost and performance in using LLMs between different languages is an important topic.
- The approach taken by the authors (i.e., the use of the FLORES-200 parallel corpus) seems sound.
- The observations made in their experiments are informative.


**Weaknesses:**

- There is little technical novelty in the work.
- The authors point out the problems caused by the disparities but do not present a concrete solution. They argue that LLMs should be trained from scratch with a multilingually fair subword tokenizer but do not provide any experimental results towards that solution. I would be interested to see how much the performance and efficiency of an LLM in English needs to be sacrificed to achieve the parity.


**Questions:**

- Would it be difficult to conduct some (preliminary) experiments regarding the proposals discussed in Section 6?


**Limitations:**

Yes

---

> ### Author Rebuttal · Authors · 2023-08-05
>
> > The authors point out the problems caused by the disparities but do not present a concrete solution. They argue that LLMs should be trained from scratch with a multilingually fair subword tokenizer but do not provide any experimental results towards that solution.
>
> Please refer to our "On the development of a multilingually fair tokenizer" comment discussing a strategy we are currently working on. We also explain why it is non-trivial to implement it in practice and why we believe it is a separate work in its own right.
>
> > I would be interested to see how much the performance and efficiency of an LLM in English needs to be sacrificed to achieve the parity.
>
> The additional cost for English would be much smaller than the total benefit for the rest of the languages. That is because increasing the vocabulary size has diminishing returns: the additional tokens correspond to increasingly rare (parts of) words. Therefore, by removing rarely used English (sub)words and replacing them with frequently used (sub)words in other languages, we would likely see an overall net befit.

---

> > ### Comment · Reviewer_Qimt · 2023-08-19
> >
> > Thank you for your response. I understand that developing a multilingually fair tokenizer is non-trivial and can be regarded as a separate piece of work, but I still think that presenting preliminary experimental results is desirable. I am not fully convinced by the authors' argument that the additional cost for English would be much smaller than the total benefit for the rest of the languages. I think there is a non-negligible possibility that the additional cost for English (or major target languages) is so large that developing a multilingually fair tokenizer is not possible in practice.

---

> > > ### Author Response · Authors · 2023-08-21
> > >
> > > Regarding your concern about the effect on the dominant language when reducing the tokens for it, perhaps you will find [this plot](https://ibb.co/7ymNkP7) interesting. It shows how many tokens would be necessary for the tokenizer of ChatGPT/GPT-4 to encode all of the English corpus of FLORES-200 for different vocabulary sizes. The result is that with __only one-third__ of the vocabulary, English sequences will become __just 10% longer.__  A 10-fold reduction in the vocabulary would result in only 30% longer sequences. English would still be treated better than how the same tokenizer treats the _cheapest_ other language, Portuguese, which is 50% longer than English. Therefore, we believe that this is not a prohibitively large cost for English.
> > >
> > > Whether such a trade-off is acceptable for a specific model or not, however, depends on the number of languages used and their similarity, and is a design choice that lies with the model developers.

---

### Official Review · Reviewer_EmCj · 2023-07-11

**Soundness:** 3 good
**Presentation:** 3 good
**Contribution:** 2 fair
**Rating:** 5
**Confidence:** 3

**Summary:**

The paper proposes the concept of tokenizer parity as a way to measure the fairness of tokenization across different languages in natural language processing. The authors argue that achieving tokenizer parity is necessary to improve the performance of multilingual models and address potential unfairness in the cost of accessing commercial language services, processing time and latency, and the amount of content that can be provided as context to the models. The paper suggests that training language models from scratch with a multilingually fair subword tokenizer is the only approach that can effectively address tokenization unfairness and achieve tokenizer parity. The authors provide several examples to demonstrate the effectiveness of the proposed method for measuring tokenizer parity and suggest that a variation of subword tokenization is necessary to achieve parity. The paper contributes to the field of natural language processing by proposing a new metric for measuring fairness in tokenization and providing guidance on how to achieve tokenizer parity in multilingual models.

**Strengths:**

In terms of originality, the paper introduces the concept of "tokenizer parity" as a systematic way to assess the fairness of tokenization across different languages. This is a novel idea that has not been explored extensively in the literature.

In terms of quality, the paper provides a detailed analysis of the tokenization lengths of different languages using various tokenizers. The authors also propose a method to measure tokenizer parity and demonstrate its effectiveness using several examples.

In terms of clarity, the paper is well-written and easy to follow. The authors provide clear definitions of the key concepts and use examples to illustrate their points.

In terms of significance, the paper addresses an important issue in natural language processing, namely the fairness of tokenization across different languages. The concept of tokenizer parity has the potential to improve the performance of multilingual models and make natural language processing more equitable across different languages. Therefore, the paper could have a positive impact on the development of multilingual models.

**Weaknesses:**

One major weakness is that the paper does not provide a clear roadmap for how the concept of tokenizer parity could be integrated into existing natural language processing pipelines. While the authors suggest that training language models with a multilingually fair subword tokenizer is the only approach that can effectively address tokenization unfairness, they do not provide guidance on how this could be achieved in practice. More detailed recommendations for how to integrate tokenizer parity into existing natural language processing pipelines would be helpful for researchers and practitioners in the field.

Another potential weakness is the lack of discussion on the impact of tokenizer disparity on downstream task accuracies. This aspect is particularly interesting and may attract more attention.

**Questions:**

1. Can you provide more detailed recommendations for how to integrate tokenizer parity into existing natural language processing pipelines?
2. What is the potential impact of tokenizer disparity on downstream task accuracies?

**Limitations:**

The lack of clear guidance on how to integrate tokenizer parity into existing natural language processing pipelines may limit the impact of the paper's findings. Additionally, the absence of discussion on the impact of tokenizer disparity on downstream task accuracies is a limitation that could be addressed in future research. Without this information, it may be difficult for researchers and practitioners to fully understand the potential benefits of using a multilingually fair subword tokenizer.

---

> ### Author Rebuttal · Authors · 2023-08-05
>
> > One major weakness is that the paper does not provide a clear roadmap for how the concept of tokenizer parity could be integrated into existing natural language processing pipelines. While the authors suggest that training language models with a multilingually fair subword tokenizer is the only approach that can effectively address tokenization unfairness, they do not provide guidance on how this could be achieved in practice.
>
> We wrote an extensive explanation about the challenges of developing a multilingually fair tokenizer in our "On the development of a multilingually fair tokenizer" comment. We hope that addresses your concerns.
>
> > 1.Can you provide more detailed recommendations for how to integrate tokenizer parity into existing natural language processing pipelines?
>
> In most language processing pipelines, tokenization happens before any other training or modelling efforts. Therefore, tokenization can be independently addressed without having to adjust other elements of the pipeline. Learning a tokenizer is typically posed as an optimization problem: finding the vocabulary that minimizes the number of tokens necessary to encode a given corpus. Obtaining a multilingually fair tokenizer can be thought of as the addition of a constraint that content in different languages should have approximately similar tokenized lengths. Please see our "On the development of a multilingually fair tokenizer" comment, where we outline one way towards implementing that.
>
> > 2.What is the potential impact of tokenizer disparity on downstream task accuracies?
>
> We intentionally focus on tokenizers to show that disparities between languages exist even before the model is trained, and even without considering the differences in the model performance. Still, tokenization does also affect downstream performance. In the Related Works section, we cite the work by Zhang et al. (2022), which shows that a balanced tokenizer corpus results in better translation performance. The works of Hofmann et al. (2021, 2022) also show that larger tokens, as well as tokenization informed by morphological structures, result in better downstream performance. We will add them to our Related Works section and will discuss further the downstream implications of the tokenization disparities.
>
> > Additionally, the absence of discussion on the impact of tokenizer disparity on downstream task accuracies is a limitation that could be addressed in future research. Without this information, it may be difficult for researchers and practitioners to fully understand the potential benefits of using a multilingually fair subword tokenizer.
>
> We respectfully disagree: even if there weren’t downstream improvements, the fundamental benefit of using multilingually fair tokenizers is ensuring that users of different languages pay the same for the same service, can process similar amounts of content, and can enjoy similar generation speeds. Nevertheless, as mentioned above, we will add a discussion highlighting the prior works showing that better tokenization does improve downstream task performance.
>
> --
>
> Zhang, Shiyue, et al. “How robust is neural machine translation to language imbalance in multilingual tokenizer training?” Biennial Conference of the Association for Machine Translation in the Americas (2022)
>
> Hofmann, Valentin, et al. “Superbizarre Is Not Superb: Derivational Morphology Improves BERT’s Interpretation of Complex Words”. Annual Meeting of the Association for Computational Linguistics (2021)
>
> Hofmann, Valentin, et al. “An Embarrassingly Simple Method to Mitigate Undesirable Properties of Pretrained Language Model Tokenizers”. Annual Meeting of the Association for Computational Linguistics (2022)

---

### Official Review · Reviewer_Fdv1 · 2023-07-28

**Soundness:** 3 good
**Presentation:** 3 good
**Contribution:** 3 good
**Rating:** 7
**Confidence:** 4

**Summary:**

The paper is very interesting in highlighting the disparities in tokenization across different languages, leading to cost, latency, and long-distance modeling inequalities in LLMs. The authors show that this is not limited to one type of tokenizer or a single family of LLMs. They show that different types of tokenizers for LLMs in practice, including subword, multilingual, or byte-level tokenization, all show disparities for languages in FLORES-200.

Based on these results, the paper convincingly argues that we need multilingually fair tokenizers for future LLMs, so that some languages are not in disadvantages in terms of the cost to access LLMs, the latency of the service, and the amount of data that can be processed.

**Strengths:**

The paper is well written and the first one (to my knowledge) to study tokenization disparities in LLMs at this scale.

The presented results will be very interesting to LLM researchers and practitioners in understanding the disparities in tokenization across different languages in different LLMs. This has major consequences for some languages, putting them at a disadvantage in terms of the cost to access LLMs, the latency of the service, and the amount of data that can be processed.



**Weaknesses:**

Section 6 argues that future tokenizers should be based on subword tokenization and support all Unicode codepoints. However, one of the main weaknesses of the paper is that it does not develop such multilingual fairer tokenizers, and compares them with existing tokenizers in practice. Attempting to do that would have emphasized challenges in developing such tokenizers and how one can overcome them.

I found the argument around disparities in long-distance modeling a bit thin. The second paragraph of Section 5.3 discusses this briefly, but additional discussion or experiments are needed to strengthen the argument. For example, it might be helpful to include analysis with multilingual document summarization (e.g., XLSum).

**Questions:**

I think the paper does a very good job of explaining why there are language disparities among different tokenizers. However, I feel these reasons are known already, what are the challenges in overcoming them?

**Limitations:**

The paper presents a large-scale study of disparities in tokenization across different languages. It discusses how LLMs are putting some languages to their disadvantage due to this. The paper itself does not propose a new tokenization scheme that needs to address its limitations.

---

> ### Author Rebuttal · Authors · 2023-08-05
>
> > I found the argument around disparities in long-distance modeling a bit thin. The second paragraph of Section 5.3 discusses this briefly, but additional discussion or experiments are needed to strengthen the argument. For example, it might be helpful to include analysis with multilingual document summarization (e.g., XLSum).
>
> The critical issue is that assuming a fixed context size, the more tokens required for the same content, the smaller the maximum size of the content that can be processed. For example, if the context of GPT-4 is just enough to fit a blog post in English, it can only fit 1/15 of the same blog post when translated into Shan. Experiments comparing the ratio content that can be represented would not bring additional value as they would be precisely the reciprocal of the premium values. Furthermore, Liu et al. (2023) show that even if the content can fit in the context size, the performance of language models decreases as the input sizes grow. We will extend Section 5.3 in terms of clarity of the writing and depth of argumentation to explain this better.
>
>
> > I think the paper does a very good job of explaining why there are language disparities among different tokenizers. However, I feel these reasons are known already, what are the challenges in overcoming them?
>
> Please check our "On the development of a multilingually fair tokenizer" comment, where we explain what are the challenges in overcoming these language disparities and what is the approach we are working on towards addressing them.
>
> --
>
> Liu, Nelson F., et al. "Lost in the middle: How language models use long contexts." arXiv preprint arXiv:2307.03172. (2023)

---

### Author Rebuttal · Authors · 2023-08-05

# On the development of a multilingually fair tokenizer

Reviewers Fdv1, EmCj and Qimt mentioned as a shortcoming of our work that we did not develop a new multilingually fair tokenizer. We would like to highlight several reasons why we found this to be more challenging than it may seem. These are all problems that we are actively working on and we hope to soon have a follow up work addressing them.

**How to account for token sharing across languages:**
As discussed in the paper, the byte-level, character-level and word-level tokenizers cannot achieve tokenization parity and subword tokenization is needed. However, simply training a subword tokenizer on a balanced dataset is also not sufficient as languages can share tokens. For example, “hotel” is written the same way in English, Spanish, Italian, Portuguese, Dutch, Danish, Hungarian, Polish and more. Hence, languages from more numerous language families will also witness shorter tokenization lengths while more isolated languages, e.g. Korean, would see larger language premiums. (The spelling of “hotel” in Korean is “호텔” and no other language has the same spelling as no other language uses the Korean script).

Instead, we suggest a two-stage process: first, training individual monolingual tokenizers for all target languages, and then merging them while maintaining parity. The merging can be done by starting with the 256 tokens corresponding to each value a byte can take and then repeatedly adding the most frequently used token for the language with the highest premium. This approach can account for the shared tokens across languages. For example, if at some stage Polish has the highest premium, adding the token for “hotel” will simultaneously reduce the premiums for all the other languages using the same spelling.

However, training 200 individual tokenizers is computationally expensive. We looked into leveraging already trained tokenizers but they are often incomparable. For example, it is not trivial to combine BPE-based tokens and SentencePiece tokens. One reason is that SentencePiece pre-processes the input by splitting it at spaces and has tokens which cannot be at the beginning of a word, something incompatible with BPE. Therefore, the only methodologically clean way we see to proceed is via training individual tokenizers from scratch using the same tokenization procedure.

**The lack of large multilingual parallel corpora and how to get around it:**
A separate challenge is the need for a large parallel corpus. FLORES-200 is rather small, just 2000 sentences. It is good enough for the evaluation of parity but not sufficient for building a fair tokeniser. That is because there might be characters and words not present in FLORES-200 which should nevertheless be present in the vocabulary. Furthermore, to do proper evaluation, a split into training and test datasets is necessary. This would reduce the training data even further. Unfortunately, we are not aware of a larger parallel corpus spanning so many languages. Constructing one is extremely difficult and expensive as well.

That is why we are looking into approximating a multilingual parallel corpus by many bilingual parallel corpuses. In such a case, the parity between, for example, Greek and Shan would not be evaluated only using Greek—Shan translations (which may not be available) but also Greek—English and English—Shan translations, Greek—German and German—Shan translations, etc. However, this requires further analysis into the conditions under which leveraging bilingual data in such a way would constitute a valid approximation.

Because of these reasons, we believe that properly building a multilingually fair tokenizer is a substantial piece of work which requires its own special treatment. We are currently working towards it but cannot possibly give it justice in the limited space of the present paper. Nevertheless, we will extend the discussion in Section 6 to better highlight these challenges.

---

### Decision · Program_Chairs · 2023-09-21

**Decision:**

Accept (poster)

**Comment:**

This paper investigates the disparities between languages caused by tokenization methods used in LLMs and highlights a potential source of unfairness in language models.

In general the reviewers liked the paper.
In terms of originality the reviewers found the paper to be the first to study tokenization disparities in LLMs.
Reviewers also liked the detailed analysis and found the work to have significant value: Addresses language fairness (EmCj, Qimt), could stimulate new research (qshy), and has practical consequences like cost and latency (Fdv1, okDv).
All reviewers found the paper to be well written.

On the other hand, reviewers commonly point out the paper's lack of practical solutions to the problem studied, as well as gaps in argumentation and exploration. Some of the more major concerns were resolved during the discussion period.